# ★STARFLOW: Scaling Latent Normalizing Flows for High-resolution Image Synthesis

**Jiatao Gu, Tianrong Chen, David Berthelot, Huangjie Zheng, Yuyang Wang,
Ruixiang Zhang, Laurent Dinh, Miguel Angel Bautista, Josh Susskind, Shuangfei Zhai**
Apple
{jgu32, szhai}@apple.com

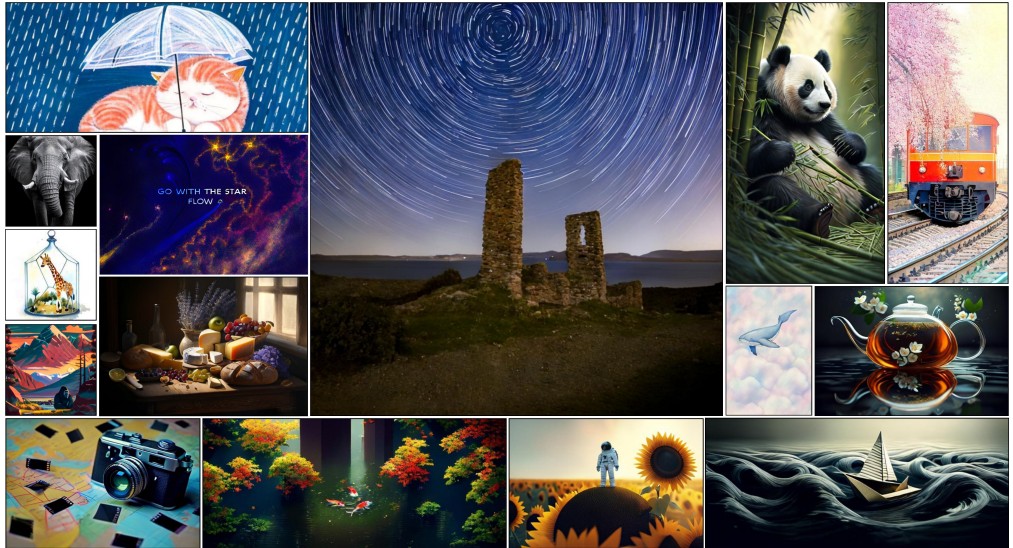

Figure 1: Text conditioned high-resolution samples of variable aspect ratios generated from our 3.8B STARFlow model. Resolutions are adjusted for the ease of visualization.

## Abstract

We present STARFlow, a scalable generative model based on normalizing flows that achieves strong performance on high-resolution image synthesis. STARFlow's main building block is Transformer Autoregressive Flow (TARFlow), which combines normalizing flows with Autoregressive Transformer architectures and has recently achieved impressive results in image modeling. In this work, we first establish the theoretical universality of TARFlow for modeling continuous distributions. Building on this foundation, we introduce a set of architectural and algorithmic innovations that significantly enhance the scalability: (1) a deep-shallow design where a deep Transformer block captures most of the model's capacity, followed by a few shallow Transformer blocks that are computationally cheap yet contribute non-negligibly, (2) learning in the latent space of pretrained autoencoders, which proves far more effective than modeling pixels directly, and (3) a novel guidance algorithm that substantially improves sample quality. Crucially, our model remains a single, end-to-end normalizing flow, allowing exact maximum likelihood training in continuous space without discretization. STARFlow achieves competitive results in both class- and text-conditional image generation, with sample quality approaching that of state-of-the-art diffusion models. To our knowledge, this is the **first** successful demonstration of normalizing flows at this scale and resolution. Code and weights available at https://github.com/apple/ml-starflow.

39th Conference on Neural Information Processing Systems (NeurIPS 2025).

# 1 Introduction

Recent years have witnessed remarkable progress in high-resolution text-to-image generative modeling, with state-of-the-art approaches predominantly falling into two distinct categories. On one hand, diffusion models (Ho et al., 2020; Rombach et al., 2022; Peebles & Xie, 2023; Esser et al., 2024) operating in continuous space have set new benchmarks in image quality. However, their reliance on iterative denoising processes renders both training and inference computationally intensive. On the other hand, autoregressive image generation methods (Yu et al., 2022; Sun et al., 2024; Tian et al., 2024)—inspired by the success of large language models (LLMs, Brown et al., 2020; Dubey et al., 2024)—avoid such inefficiencies by modeling images in discrete space via quantization; yet, this quantization can impose stringent limitations and adversely affect fidelity. More recently, a promising trend has emerged to explore hybrid models (Li et al., 2024; Gu et al., 2024b; Fan et al., 2024) that apply autoregressive techniques directly in continuous space. However, the inherently distinct characteristics of these two paradigms introduce additional complexity in effective unification.

In this paper, we turn our eyes on the yet another modeling approach – Normalizing Flows (NFs, Rezende & Mohamed, 2015; Dinh et al., 2016), a family of likelihood based models that have received relatively little attention in the recent wave of Generative AI. We start from inspecting TARFlow (Zhai et al., 2024), a recently proposed model that combines a powerful Transformer architecture with autoregressive flows (AFs, Kingma et al., 2016; Papamakarios et al., 2017). While TARFlow demonstrates promising results on the potential of NFs as a modeling principle, it remains unclear whether it can perform as a scalable method, in comparison to other approaches such as diffusion and discrete autoregressive models. To this end, we propose STARFlow, a family of generative models that shows for the **first-time** that NF models can successfully generalize to high-resolution and large-scale image modeling. We first provide a theoretical insight on *why AFs can be capable generative models* by showing the universality of multi-block AFs in modeling continuous distributions. On top of this, we propose a novel *deep–shallow* architecture. We found that the architecture configuration, e.g., the number of flows as well as the depth and width of the Transformer for each flow, plays a pivotal role to the model's performance. While TARFlow (Zhai et al., 2024) proposes to uniformly allocate model depth among all flows, we found that it is beneficial to have a skewed architecture design, where we allocate most of the model parameters to the first AF block (i.e., the one closest to the prior), which is followed by a few shallow but non-negligible blocks. Importantly, our model still yields a stand-alone normalizing-flow framework that supports end-to-end maximum-likelihood training in continuous space, thereby sidestepping the quantization limits inherent to discrete models. Rather than operating directly in data space, we instead learn AFs in the latent space of pretrained autoencoders. Crucially, we demonstrate that NFs align naturally with compressed latents—an intuitive yet vital observation—enabling far superior modeling of high-resolution inputs, as verified in our experiments, compared with training directly on pixels. Similar to TARFlow, noise injection proves essential: by fine-tuning the decoder, we train the model on noisy latents and at the same time simplify the original sampling pipeline. Moreover, we revisit the classifier-free guidance (CFG) algorithm for AFs from a more principled way and propose a novel guidance algorithm, which substantially improves image quality, especially at high guidance weights in text-to-image generation tasks.

Together, these innovations represent the first demonstration of NF models applied to large-scale, high-resolution image generation. Our approach offers a scalable and efficient alternative to conventional diffusion-based and autoregressive approaches, achieving competitive performance on benchmarks for both class-conditioned image and large-scale text-to-image synthesis. Moreover, our framework is highly flexible, and we demonstrate that it easily enables interesting settings such as image inpainting and instruction based image editing by finetuning.

# 2 Preliminaries

## 2.1 Normalizing Flows

In this paper, we consider Normalizing Flows (NFs, Rezende & Mohamed, 2015; Dinh et al., 2014, 2016) as the class of likelihood method that follows the change of variable formula. Given continuous inputs $x \sim p_{\text{data}}$, $x \in \mathbb{R}^D$, a NF learns an invertible transformation $f_\theta : \mathbb{R}^D \mapsto \mathbb{R}^D$ (with $\theta$ being the parameters) which maps data $x$ into the noise space $f_\theta(x)$, and can be trained with maximum

likelihood estimation (MLE):

$$\max_{\theta} \ \mathbb{E}_{\boldsymbol{x} \sim p_{\text{data}}} \log p_{\text{NF}}(\boldsymbol{x}; \theta) = \log p_0(f_\theta(\boldsymbol{x}); \theta) + \log\left(\left|\det\left(\frac{\partial f_\theta(\boldsymbol{x})}{\partial \boldsymbol{x}}\right)\right|\right), \qquad (1)$$

where the first term rewards sending data to high-density regions of the prior $p_0$, while the Jacobian term penalizes excessive local volume shrinkage, ensuring the transformation remains bijective and does not collapse nearby points onto a lower-dimensional set. One automatically obtains a generative model by inverting $f_\theta$, with a sampling procedure $\boldsymbol{z} \sim p_0(\boldsymbol{z})$, $\boldsymbol{x} = f_\theta^{-1}(\boldsymbol{z})$.

## 2.2 Autoregressive Flows and TARFlow

An interesting variant of NFs is autoregressive flows (AFs, Kingma et al., 2016; Papamakarios et al., 2017). In the simplest affine form, an AF constructs $\boldsymbol{z} = f_\theta(\boldsymbol{x}) = \{\mu_\theta, \sigma_\theta\}(\boldsymbol{x})$ as a standalone invertible model with the forward ($\boldsymbol{x} \rightarrow \boldsymbol{z}$) and sampling ($\boldsymbol{z} \rightarrow \boldsymbol{x}$) process:

$$\boldsymbol{z}_d = \left(\boldsymbol{x}_d - \mu_\theta(\boldsymbol{x}_{<d})\right)/\sigma_\theta(\boldsymbol{x}_{<d}), \qquad \boldsymbol{x}_d = \mu_\theta(\boldsymbol{x}_{<d}) + \sigma_\theta(\boldsymbol{x}_{<d}) \cdot \boldsymbol{z}_d, \ \ \forall d \in [1, D], \qquad (2)$$

where $\boldsymbol{x}_0$ is a constant `<sos>`. This can be seen as "next-token prediction" with affine transformation, and training with Eq. (1) where the Jacobian term becomes extremely simple as $-\sum_{d=1}^{D} \log \sigma_\theta(\boldsymbol{x}_{<d})$. The extension to multi-channel inputs $\boldsymbol{x} \in \mathbb{R}^{D \times C}$ (e.g., $C = 3$ for RGB image) is immediate as channels at each step can be treated as conditionally independent. We *omit* the channel dim henceforth.

Recently, Zhai et al. (2024) introduced TARFlow, a compelling framework for building performant NFs for image data. Specifically, TARFlows can be viewed as a special form of AFs by pairing causal-Transformer blocks with an extension of classical AF formulation – stacking multiple AF layers whose autoregressive ordering **alternates** from one layer to the next. To be concrete, with $T$ flows, we have $\boldsymbol{z} = f_\theta^T \circ f_\theta^2 \circ \cdots \circ f_\theta^1(\boldsymbol{x})$, where each block $f_\theta^t(.)$ processes the input in its own ordering $\boldsymbol{x}_\pi = (\boldsymbol{x}_{\pi_1}, \ldots, \boldsymbol{x}_{\pi_D})$ (a permutation of $\{\boldsymbol{x}_1 \ldots \boldsymbol{x}_D\}$), enabling the stack to capture dependencies in both directions of the data sequence. Training is still performed end-to-end:

$$\max_{\theta} \ \mathbb{E}_{\boldsymbol{x} \sim p_{\text{data}}} \log p_{\text{AF}}(\boldsymbol{x}; \theta) = -\frac{1}{2}\|\boldsymbol{z}\|_2^2 - \sum_{t=1}^{T} \sum_{d=1}^{D} \log \sigma_\theta^t(\boldsymbol{x}_{\pi_{<d}}^t), \qquad (3)$$

where $\boldsymbol{x}^t = f_\theta^t(\boldsymbol{x}^{t-1})$ defines the forward propagation (Eq. (2)); we denote the data $\boldsymbol{x} = \boldsymbol{x}^0$ and the final output $\boldsymbol{z} = \boldsymbol{x}^T$ is modeled with standard Gaussian. Additionally, Zhai et al. (2024) also proposed several techniques to improve the modeling capability, including noise augmented training, score-based denoising and incorporating guidance (Ho & Salimans, 2021).

# 3 STARFlow

In this section, we propose $\underline{\text{S}}$calable $\underline{\text{T}}$ransformer $\underline{\text{A}}$uto$\underline{\text{r}}$egressive $\underline{\text{Flow}}$ (STARFlow), a method that pushes the frontier of NF based high-resolution image generation. We first establish—on theoretical grounds—AFs' expressivity as a general modeling method in § 3.1, based on which we propose our core approaches by improving TARFlow in several key aspects: (1) a better architecture configuration (§ 3.2), (2) a working recipe of learning in the latent space (§ 3.3) and (3) a novel guidance algorithm (§ 3.4). An illustration of the learning and inference pipeline is presented in Fig. 4.

## 3.1 Why TARFlows are Capable Generative Models?

While empirical results confirm that TARFlow is highly competitive (Zhai et al., 2024), we ask—from a modeling perspective—whether they are expressive enough to warrant scaling. Here, we claim:

> **Proposition 1.** Stacked autoregressive flows with $T \geq 2$ blocks of $D$ autoregressive steps and alternating orderings are *highly expressive* for modeling continuous densities on $\mathbb{R}^D$.

*Sketch of Proof.* Let's consider $T = 2$. Without loss of generality, we model $f_\theta = f_\theta^a \circ f_\theta^b$ where $f_\theta^a$ and $f_\theta^b$ employ reversed orderings (forward and backward) for data $\boldsymbol{x} \in \mathbb{R}^D$:

$$\boldsymbol{x}_d = \mu_\theta^b(\boldsymbol{x}_{<d}) + \sigma_\theta^b(\boldsymbol{x}_{<d}) \cdot \boldsymbol{y}_d, \ \ \boldsymbol{y}_d = \mu_\theta^a(\boldsymbol{y}_{>d}) + \sigma_\theta^a(\boldsymbol{y}_{>d}) \cdot \boldsymbol{z}_d, \ \ \boldsymbol{z}_d \sim \mathcal{N}(\boldsymbol{0}, I), \ \ d \in [1, D]. \ \ (4)$$

We assume $\boldsymbol{z} \sim \mathcal{N}(0, I)$ under the *base* distribution. This yields the autoregressive factorization $p(\boldsymbol{x}) = \prod_{d=1}^{D} p(\boldsymbol{x}_d \mid \boldsymbol{x}_{<d})$ as follows:

$$p(\boldsymbol{x}_d \mid \boldsymbol{x}_{<d}) = \int \mathcal{N}\big(\boldsymbol{x}_d \mid \hat{\mu}_\theta(\boldsymbol{x}_{<d}, \boldsymbol{y}_{>d}), \hat{\sigma}_\theta^2(\boldsymbol{x}_{<d}, \boldsymbol{y}_{>d})I\big) \cdot p(\boldsymbol{y}_{>d} \mid \boldsymbol{x}_{<d})\mathrm{d}\boldsymbol{y}_{>d}, \tag{5}$$

where $\hat{\mu}_\theta = \mu_\theta^b(\boldsymbol{x}_{<d}) + \mu_\theta^a(\boldsymbol{y}_{>d})\sigma_\theta^b(\boldsymbol{x}_{<d})$, $\hat{\sigma}_\theta = \sigma_\theta^a(\boldsymbol{y}_{>d})\sigma_\theta^b(\boldsymbol{x}_{<d})$ defined in Eq. (4).

For every $d < D$, we have $\boldsymbol{y}_{>d} \neq \varnothing$, so Eq. (5) is a latent-variable marginalization of Gaussians (mixture-like) and can represent complex continuous conditionals. For the final coordinate $d = D$ we have $\boldsymbol{y}_{>D} = \varnothing$. Eq. (5) reduces to a single Gaussian and the expressivity is restricted. This restriction can be lifted by extending additional augmented variables. Moreover, the above derivation only uses the *base* assumption $\boldsymbol{z}_d \sim \mathcal{N}(0, 1)$ in the generative direction. In general, conditioning on observed coordinates induces a non-Gaussian latent distribution (i.e., $q_\theta(\boldsymbol{z}_d \mid \boldsymbol{x}_{<d})$ is not necessarily Gaussian). Consequently, the resulting con-

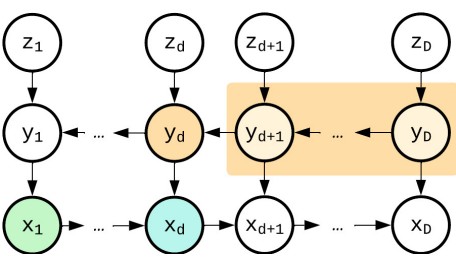

Figure 2: An illustration of 2-block AFs.

ditional $q_\theta(\boldsymbol{x}_D \mid \boldsymbol{x}_{<D})$ can be even more complex than a single Gaussian. Additional derivation details appear in the Appendix A. □

The preceding proposition clarifies why we can safely scale-up AFs on large data. Even in the minimal setting $T = 2$ where full universality is not attained, the resulting limitation is negligible in high-dimensional domains such as natural images.

## 3.2 Proposed Architecture

The derivation in § 3.1 motivates a redesign of scalable AF architectures within realistic computational budgets, emphasizing that we need not greatly expand the number of flow blocks—indeed (even $T = 2$ often suffices). However, the remark leaves unresolved how best to allocate compute across those blocks. We first inspect the proposed architecture configuration in TARFlow, which suggests to allocate equal sized Transformer layers for each flow. Interestingly, in our reproduced TARFlow results, we see that most effective compute (measured through the lens of guidance) concentrates in just the top few AF blocks

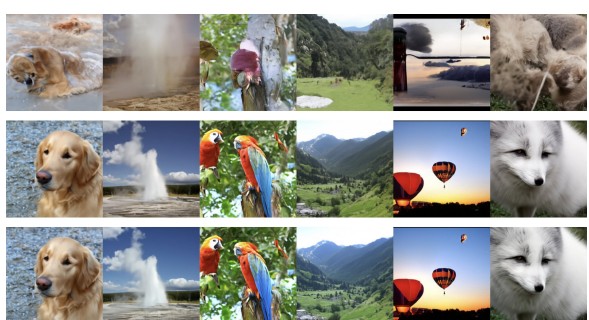

Figure 3: Top to bottom, guiding the first 0, 3, 8 flow blocks with a TARFlow model with 8 flow blocks. We see that guidance is only effective up to the top 3 blocks.

(see motivating examples Fig. 3). We conjecture that end-to-end training drives the network to exploit layers closest to the noise, a behavior that contrasts that of diffusion models.

**Deep-shallow Architecture** Our architecture can be intuitively considered as an extension of standard autoregressive language models (e.g., LLaMA (Dubey et al., 2024)) with a general *deep-shallow* design. At inference time, a deep AF block first autoregressively generates $\boldsymbol{x}^1$ from noise $\boldsymbol{z}$, followed by a sequence of shallow AF blocks that iteratively refine it to $\boldsymbol{x}^N$, all while keeping the total number of blocks $T$ small. Given a total depth budget $L$, we instantiate the model as $l(T)$: one deep $l$-layer block and $T{-}1$ shallow 2-layer blocks, satisfying $L = l + 2(T{-}1)$. This asymmetric design turns the deep block into a *Gaussian language model*, while the shallow stack plays the role of a *learned image tokenizer*.

**Conditional STARFlow** This design naturally extends to conditional generation by simply prepending the control signal (e.g., class label, caption) to the input of the flow. Interestingly, our preliminary experiments show that conditioning only the deep block—while leaving the shallow blocks to focus solely on local image refinement—incurs no loss in performance. This not only simplifies the overall architecture, but also enables seamless initialization of the deep block with any pre-trained language

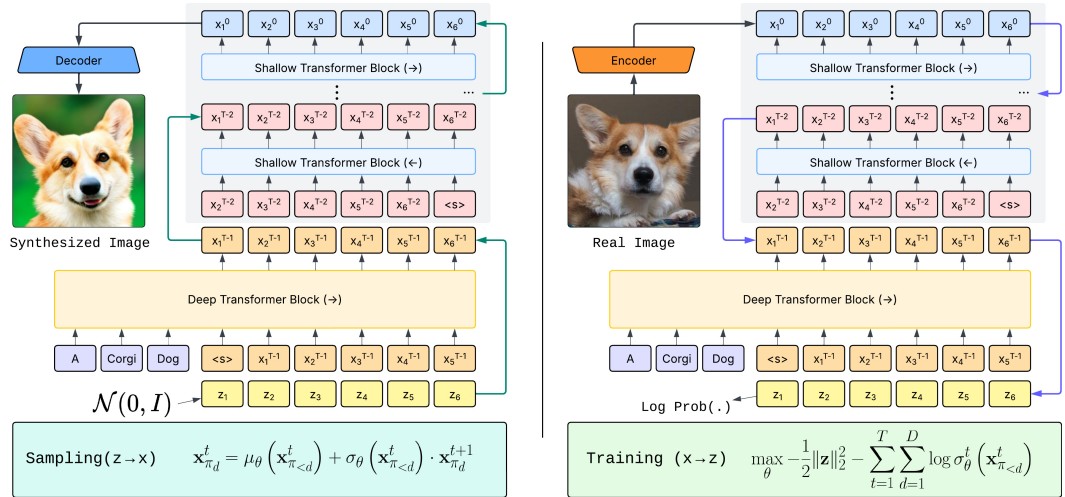

Figure 4: An illustration of the **autoregressive** inference (left) and **parallel** training (right) process of our proposed model for text-to-image generation. The upward (green) and downward (purple) arrows refers to the inverse and forward AF step as shown in Eq. (2).

model, without major modifications. As a result, our image generator can be directly integrated into any LLM's semantic space, eliminating the need for a separate text encoder.

### 3.3 Moving to Latent Space

Analogy to Stable Diffusion (SD, Rombach et al., 2022) w.r.t standard diffusion models, STARFlow directly models the latent space of a pretrained autoencoders $x \approx \mathcal{D}(\tilde{x}), \tilde{x} = \mathcal{E}(x)$, enabling high-resolution image generation. For instance, when using SD-1.4 autoencoder[1], one can reduce input shape from $256 \times 256$ to $32 \times 32$. As noted by Zhai et al. (2024), injecting a proper amount of Gaussian noise, instead of small dequantization noise Dinh et al. (2016); Ho et al. (2019), is crucial for stable training and high quality sampling. This then makes it necessary to perform an additional score-based denoising step to clean up the noise components in the samples Zhai et al. (2024).

In the context of latent normalizing flows, however, the added noise becomes an integral component of the latent representation. Specifically, we encode each sample as $\tilde{x} \sim q_{\text{enc}} = \mathcal{N}\left(\mathcal{E}(x); \sigma_L^2 I\right)$. We perform preliminary search for the noise scale ($\sigma_L$) to based on the choice of autoencoders. For example, we set $\sigma_L = 0.3$ throughout the paper.

**Learning** Learning in the latent space leaves additional flexibility that the flow model can focus on high-level semantics and leave the low-level local details with the pixel decoder. In this way, AF acts as a learnable prior for the latents. Following VAEs (Kingma & Welling, 2013), we optimize the entire model by maximizing the evidence lower-bound (ELBO) where the entropy term is constant:

$$\max_{\theta,\phi} \mathbb{E}_{\tilde{x} \sim q_{\text{enc}}(\tilde{x}|x), x \sim p_{\text{data}}} \left[\log p_{\text{AF}}(\tilde{x}; \theta) + \log p_{\text{dec}}(x|\tilde{x}; \phi) - \log q_{\text{enc}}(\tilde{x}|x)\right], \qquad (6)$$

where $\phi$ are the parameters of decoder $p_{\text{dec}}$ which transforms the noisy latents back to the pixel space. Here, we jointly train the AF prior and pixel decoder, freezing the encoder distribution –as in SD–, which stabilizes training and decouples their optimization. Relaxing the encoder $q_{\text{enc}}$ and training with the full ELBO loss including entropy regularization are left for future work.

**Pixel Decoder** As shown in Eq. (6), the prertaiend decoder has to be adapted in order to decode from the noisy latents. Different from Zhai et al. (2024) which relies on gradient-based denoising, modeling in the latent allows a simpler solution by directly fine-tuning the decoder over noisy latents:

$$\min_{\phi} \mathcal{L}\left(\mathcal{D}(\mathcal{E}(x + \sigma\epsilon); \phi), x\right), \qquad (7)$$

where following Esser et al. (2021), $\mathcal{L} = \mathcal{L}_{\text{L2}} + \mathcal{L}_{\text{LPIPS}} + \beta\mathcal{L}_{\text{GAN}}$. We empirically observe consistently better performance than score-based denoising technique proposed in (Zhai et al., 2024), with FID decreasing from 2.96 to 2.40 on ImageNet-256. See Appendix C for more discussions.

---

[1] https://huggingface.co/stabilityai/sd-vae-ft-mse.

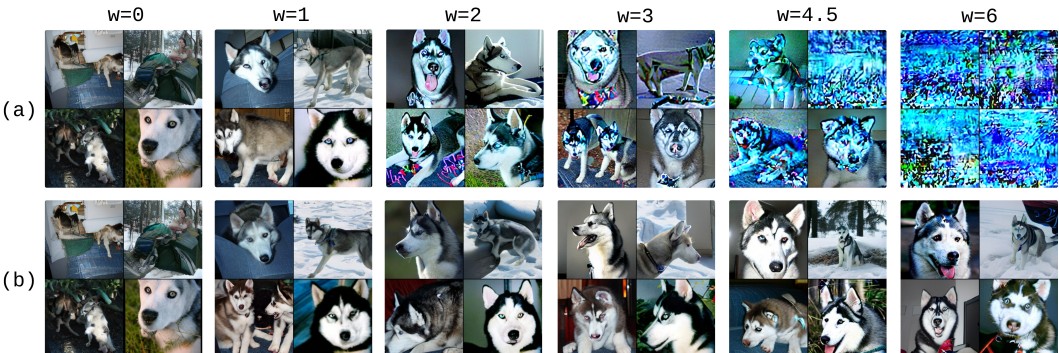

Figure 5: (a) Guidance from TARFlow (Zhai et al., 2024) (b) Proposed guidance on ImageNet $256 \times 256$.

## 3.4 Revisiting Classifier-Free Guidance for Autoregressive Flows

Classifier-free guidance (CFG), originally introduced for diffusion models (Ho & Salimans, 2021), has become a cornerstone in modern generative modeling, proving broadly effective across various architectures, including AR models (Yu et al., 2022). At a high level, CFG amplifies the difference between conditional and unconditional predictions, encouraging more mode-seeking behavior.

In the context of AFs, Zhai et al. (2024) made the first attempt to apply CFG by linearly extrapolating the mean and variance at each step (Eq. (2)): $\tilde{\mu}_c = \mu_c + \omega(\mu_c - \mu_u)$ and $\tilde{\sigma}_c = \sigma_c + \omega(\sigma_c - \sigma_u)^2$,

where $\omega > 0$ denotes the guidance weight. While effective to some extent, this naïve formulation lacks principled justification, leaving unclear how $\mu$ and $\sigma$ should be jointly modulated under guidance. Furthermore, as shown in Fig. 5, this approach becomes unstable at high guidance weights—precisely the regime required for visually compelling results in text-to-image generation.

We propose to revisit CFG from the perspective of score function, the original intuition of Ho & Salimans (2021). In short, we want to sample from a guided distribution $\tilde{p}$ which score satisfies:

$$\nabla_{\boldsymbol{x}} \log \tilde{p}_c(\boldsymbol{x}) = \nabla_{\boldsymbol{x}} \log p_c(\boldsymbol{x}) + \omega \left( \nabla_{\boldsymbol{x}} \log p_c(\boldsymbol{x}) - \nabla_{\boldsymbol{x}} \log p_u(\boldsymbol{x}) \right). \tag{8}$$

It is generally non-trivial to determine $\tilde{p}_c$ for every flow block. Fortunately, under the design of our proposed model, guidance is only required in the deep block, which functions as a *Gaussian Language Model* (§ 3.2). Therefore, Eq. (8) can be easily simplified into the following:

---

**Proposition 2.** Given $p_u = \mathcal{N}(\mu_u, \sigma_u^2 I)$, and $p_c = \mathcal{N}(\mu_c, \sigma_c^2 I)$, the guided distribution $\tilde{p}_c$ is also Gaussian $\tilde{p}_c = \mathcal{N}(\tilde{\mu}_c, \tilde{\sigma}_c^2 I)$ and satisfies:

$$\tilde{\mu}_c = \mu_c + \frac{\omega s}{1 + \omega - \omega s} \cdot (\mu_c - \mu_u), \quad \tilde{\sigma}_c = \frac{1}{\sqrt{1 + \omega - \omega s}} \cdot \sigma_c, \tag{9}$$

where $s = \sigma_c^2 / \sigma_u^2$ and $\omega > 0$.

---

*proof*: A detailed derivation is provided in the Appendix A. □

Notably, when $\sigma_c = \sigma_u$, Eq. (9) reduces to the standard CFG used in diffusion models. However, directly applying Eq. (9) can lead to severe numerical instability, as the denominator $1 + \omega - \omega s$ may approach zero or even become negative. To address this, we propose clipping $s$ via $s = \text{CLIP}(s, 0, 1)$, motivated by the intuition that the guided distribution should be more *mode-seeking* than the original, implying that $1 + \omega - \omega s \geq 1$ for any $\omega$, therefore $s \leq 1$.

## 3.5 Applications

STARFlow is a versatile generative model that not only produces diverse, high-quality images under various conditions but also extends naturally to downstream applications. We showcase two examples: image inpainting and editing.

---

[2]We use $c$ and $u$ to denote the conditional and unconditional predictions, respectively.

**Training-Free Inpainting** We first map the masked image to the latent space, replacing masked regions with Gaussian noise. Reverse sampling is then performed, restoring unmasked pixels with ground truth. We perform generation iteratively until the final inpainted output.

**Interactive Generation and Editing** We finetune STARFlow on an image editing dataset (Fig. 6b), enabling joint modeling of generation and editing with a single conditional AF model. Its invertibility also allows direct image encoding, making it suitable for interactive use.

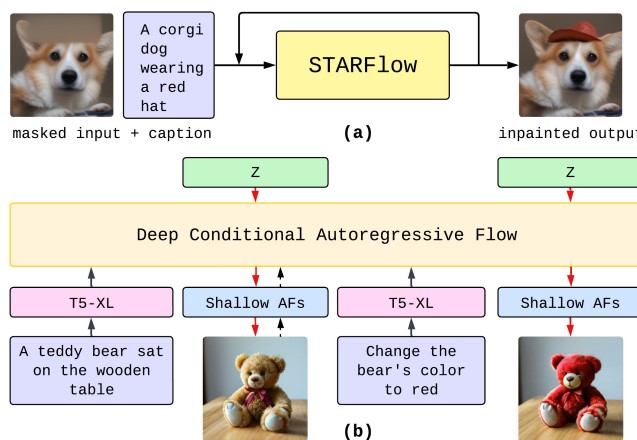

Figure 6: (a) image inpainting (b) interactive editing.

# 4 Experiments

## 4.1 Experimental Settings

**Dataset** We experiment with STARFlow on both class-conditioned and text-to-image generation tasks. For the former, we conduct experiments on ImageNet-1K (Deng et al., 2009) including $256 \times 256$ and $512 \times 512$ resolutions. For text-to-image, we show two settings: a constrained setting CC12M (Changpinyo et al., 2021), where each image is accompanied by a synthetic caption following (Gu et al., 2024a). We also demonstrated a scaled setting where our models trained an in-house dataset with CC12M, in total $\sim 700M$ text-image pairs.

**Evaluation** In line with prior works, we report Fréchet Inception Distance (FID) (Heusel et al., 2017) to quantify the the realism and diversity of generated images. For text-to-image generation, we use MSCOCO 2017 (Lin et al., 2014) validation set to assess the zero-shot capabilities of these models. We also report additional evaluation (e.g., GenEval (Ghosh et al., 2023)) in Appendix C.

**Model and Training Details** We implement all models following the setup of Dubey et al. (2024), using RoPE (Su et al., 2024) for positional encoding. By default, we set the architecture to $d(N) = 18(6)$ with a model dimension of 2048 (XL) and $24(6)$ with a dimension of 3096 (XXL) for class-conditioned and text-to-image models, respectively (§ 3.2), resulting in 1.4B and 3.8B parameters. Since STARFlow operates in a compressed latent space, we are able to train all models with a patch size of $p = 1$. For text-to-image models, we use T5-XL (Raffel et al., 2020) as the text encoder. To showcase the generality of our approach, we also train a variant where the deep block is initialized from a pretrained LLM (Gemma2 (Team et al., 2024) in this case), without additional text encoder.

All models are pre-trained at $256 \times 256$ resolution on 400M images with a global batch size of 512. High-resolution finetuning is done by increasing input length. For text-to-image models, variable-length inputs are supported via mixed-resolution training: images are pre-classified into 9 shape buckets and flattened into sequences for unified processing. See Appendix B for detailed settings.

## 4.2 Results

**Comparison with Baselines** We benchmark our approach on class-conditioned ImageNet-256, comparing against diffusion and autoregressive models across both discrete and continuous domains (Table 1). For fair comparison, we train a TARFlow model Zhai et al. (2024) in pixel space with a similar parameter count and original architecture (8 flows, 8 layers each, width 1280). We also train a variant with our *deep-shallow* design, identical to STARFlow except for using pixel inputs with linearly scaled patch sizes. Among NF models, the *deep-shallow* architecture consistently outperforms the standard design, and switching to latent-space inputs yields further gains. Our method achieves competitive results compared to other baselines (Tables 1 and 2). Note the FID on ImageNet $256 \times 256$ is near saturated to the upper-bound of the finetuned decoder (see additional details in Appendix B). Zero-shot evaluations on COCO (Table 3) show strong performance on

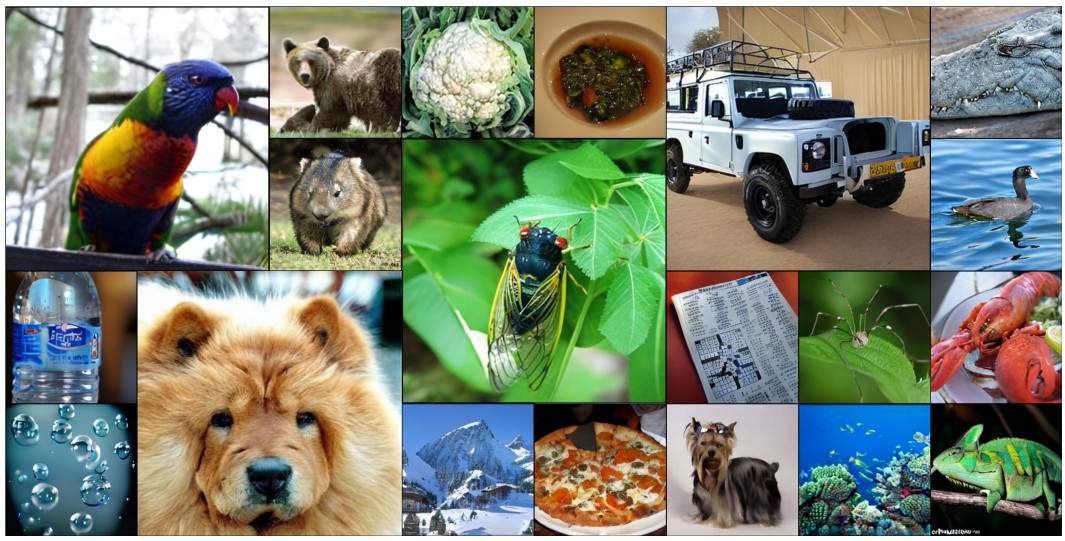

Figure 7: Random samples of STARFlow on ImageNet $256 \times 256$ and $512 \times 512$ ($\omega = 3.0$).

Table 1: Class-cond ImageNet $256 \times 256$ (FID-50K)

| Model | FID↓ | # Param. |
|---|---|---|
| **Diffusion Models** | | |
| ADM (Dhariwal & Nichol, 2021) | 10.94 | 554M |
| CDM (Ho et al., 2022b) | 4.88 | – |
| LDM (Rombach et al., 2022) | 3.60 | 400M |
| RIN (Jabri et al., 2022) | 3.76 | 410M |
| DiT (Peebles & Xie, 2023) | 2.27 | 675M |
| SiT (Ma et al., 2024) | 2.06 | 675M |
| **Autoreg. (discrete)** | | |
| VQGAN (Esser et al., 2021) | 15.78 | 1.4B |
| RQTran (Lee et al., 2022) | 3.80 | 3.8B |
| LlamaGen-3B (Sun et al., 2024) | 2.18 | 3.1B |
| VAR (Tian et al., 2024) | 1.73 | 2.0B |
| **Autoreg. (continuous)** | | |
| Jetformer (Tschannen et al., 2024b) | 6.64 | 2.75B |
| MAR-AR (Li et al., 2024) | 4.69 | 479M |
| MAR (Li et al., 2024) | 1.55 | 943M |
| DART (Gu et al., 2024b) | 3.82 | 820M |
| GIVT (Tschannen et al., 2024a) | 2.59 | – |
| **Normalizing Flow** | | |
| TARFlow (Zhai et al., 2024) [a] | 5.56 | 1.3B |
| TARFlow + *deep-shallow* | 4.69 | 1.4B |
| STARFlow (Ours) | 2.40 | 1.4B |

[a] Implemented using their official codebase.

Table 2: Class-cond ImageNet $512 \times 512$ (FID-50K)

| Model | FID↓ | # Param. |
|---|---|---|
| ADM-U (Dhariwal & Nichol, 2021) | 3.85 | 731M |
| DiT-XL/2 (Peebles & Xie, 2023) | 3.04 | 674M |
| LEGO (Zheng et al., 2024b) | 3.74 | 681M |
| MaskDiT-G (Zheng et al., 2024a) | 2.50 | 730M |
| EDM2-XXL (Karras et al., 2024) | 1.25 | 1.5B |
| STARFlow (Ours) | 3.00 | 1.4B |

Table 3: Zero-shot T2I on COCO (FID-30K)

| Method | FID↓ | # Param. |
|---|---|---|
| DALL·E (Ramesh et al., 2021) | 27.5 | 12B |
| CogView2 (Ding et al., 2021) | 24.0 | 6B |
| Make-A-Scene (Gafni et al., 2022) | 11.8 | – |
| DART (Gu et al., 2024b) | 11.1 | 800M |
| DALL·E 2 (Ramesh et al., 2022) | 10.4 | 5.5B |
| GigaGAN (Kang et al., 2023) | 9.1 | 1B |
| Muse (Chang et al., 2023) | 7.9 | 3B |
| Imagen (Ho et al., 2022a) | 7.3 | 3B |
| Parti-20B (Yu et al., 2022) | 7.2 | 20B |
| eDiff-I (Balaji et al., 2022) | 7.0 | 9B |
| STARFlow-CC12M | 10.3 | 3.8B |
| STARFlow-CC12M-Gemma | 11.4 | 2.4B |
| STARFlow-FullData | 9.1 | 3.8B |

text-conditioned generation, demonstrating that NFs can also serve as a scalable and competitive generative modeling framework.

**Qualitative Results** Fig. 7 and Appendix Fig. 10 present representative class- and text-conditioned generations, respectively. Our method delivers high-resolution images over a wide range of aspect ratios, with perceptual quality comparable to state-of-the-art diffusion and autoregressive approaches. Fig. 9 also highlights our model's support for **image editing**. Further qualitative and interactive editing results appear in Appendix G, underscoring the breadth and fidelity of our outputs.

**Comparison with Diffusion and Autoregressive Models** We further compare STARFlow with diffusion and autoregressive (AR) models to analyze training dynamics. Fig. 8a shows FID trajectories

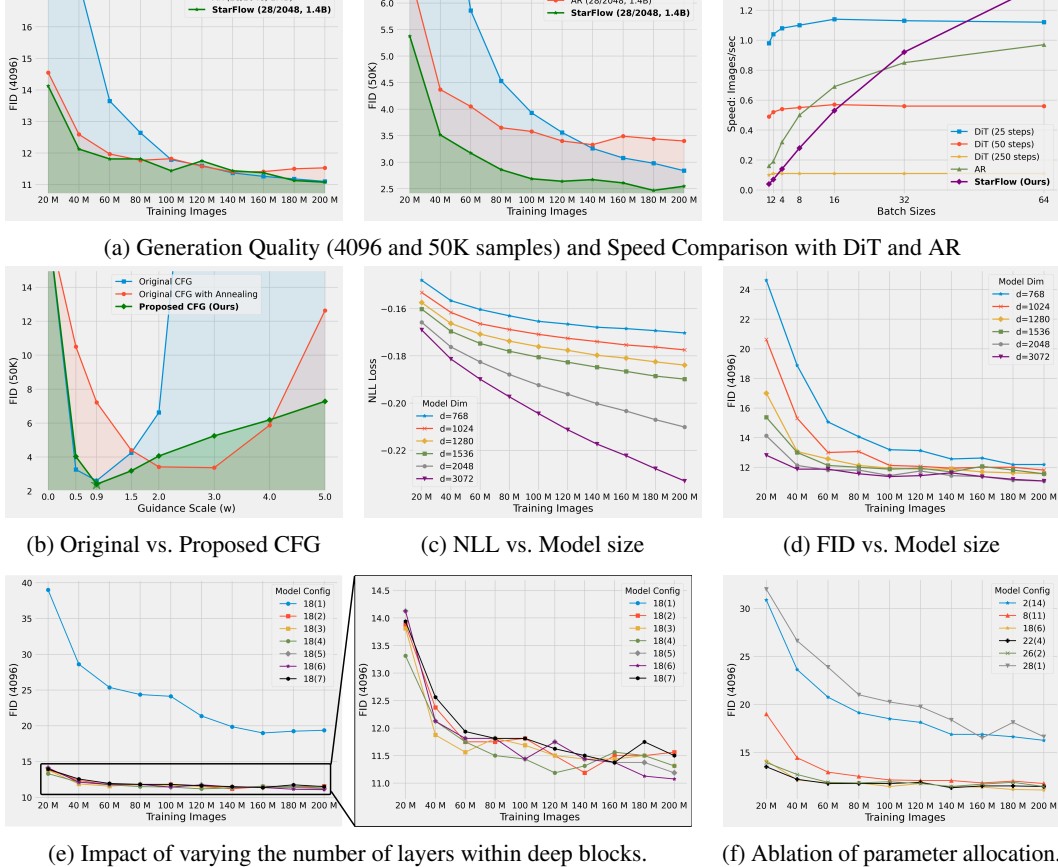

(a) Generation Quality (4096 and 50K samples) and Speed Comparison with DiT and AR

(b) Original vs. Proposed CFG    (c) NLL vs. Model size    (d) FID vs. Model size

(e) Impact of varying the number of layers within deep blocks.    (f) Ablation of parameter allocation.

Figure 8: Experimental results of comprehensive ablation study

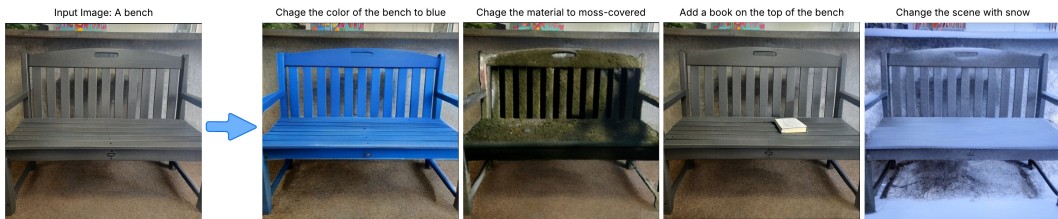

Figure 9: Example of Image editing using STARFlow. Given an input image and simple description, our model can seamlessly edit the contents based on various instruction using with the learned model prior.

using nearly identical architectures. While the FID gap between STARFlow and the baselines is smaller when computed over 4,096 samples, STARFlow consistently achieves the lowest FID at every training checkpoint when evaluated with 50,000 samples. This suggests that STARFlow produces more diverse outputs, which may not be fully captured with smaller evaluation sets.

Fig. 8a also compares inference throughput on a single H100 GPU for diffusion, AR, and STARFlow models. Diffusion's wall-clock time grows linearly with its number of refinement steps—≈ 250 steps at best FID—so it's the slowest. By contrast, each step in AR and STARFlow is only a lightweight forward pass whose per-token cost is low, allowing throughput to rise as batch size increases. Beyond a batch size of 32, STARFlow outperforms the AR baseline by restricting guidance to the deep block and removing the per-token multinomial sampling loop, yielding superior inference-time scalability.

**Comparison of CFG Strategies**    As shown in the Fig. 8b, the original strategy used in Zhai et al. (2024) exhibits a sharp "dip-and-spike" behavior: it achieves its best FID at similar guidance weight as the newly proposed CFG, but then degrades quickly as you move away from that optimum. Even when using the "annealing trick" (Zhai et al., 2024), performance still suffers dramatically both

scales. By contrast, our proposed CFG not only improves on the original's best point—without additional tricks—but—more importantly—maintains nearly the same quality over a much wider range of guidance weights, which gives more flexibility in tuning text-conditioned generation tasks.

**Scalability Analysis**    To assess the scalability, we perform a study by varying the depth of the deep block and tracking performance over training. Fig. 8c reports negative log-likelihood (NLL) and Fig. 8d shows FID with 4096 samples across iterations. Both metrics indicate that deeper models converge faster and achieve better final performance, demonstrating the increased capacity.

**Ablation on Model Design**    To validate the theoretical insights from Prop. 1, we study how model expressivity varies with the number of layers $T$ in the deep block. Performance drops sharply when $T < 2$, while models with $T \geq 2$ perform similarly—consistent with Prop. 1. We also ablate the number and depth of deep blocks in Figs. 8e and 8f, finding that block depth is more critical than quantity, providing practical guidance for architectural design.

## 5  Related Work

**Continuous Normalizing Flows, Flow Matching, and Diffusion Models**    Normalizing Flows (NFs) can be extended to continuous-time via Continuous Normalizing Flows (CNFs) (Chen et al., 2018), which model transformations as ODEs. This relaxes the need for explicit invertible mappings and simplifies Jacobian computation to a trace (Grathwohl et al., 2018), though it requires noisy stochastic estimators (Hutchinson, 1989). Flow Matching (Lipman et al., 2023), inspired by CNFs, learns sample-wise interpolations between prior and data using vector fields grounded in Tweedie's Lemma (Efron, 2011). While CNFs and NFs optimize exact likelihoods through invertible mappings, Flow Matching aligns more closely with diffusion models, sharing variational training objectives.

**Autoregressive Models**    Discrete autoregressive models, especially large language models (Brown et al., 2020; Dubey et al., 2024; Guo et al., 2025), dominate modern generative AI by scaling next-token prediction. Scaling laws (Kaplan et al., 2020) show predictable gains with more data and parameters. These models now power leading multimodal systems for both understanding and generation (Liang et al., 2024; Sun et al., 2024; Tian et al., 2024; Li et al., 2025).

To overcome information loss from quantization, recent work extends AR modeling to continuous spaces, using mixture-of-Gaussians (Tschannen et al., 2024a,b) or diffusion decoding (Li et al., 2024; Gu et al., 2024b; Fan et al., 2024). Hybrid approaches also emerge, unifying AR and diffusion paradigms (Gu et al., 2024a; Zhou et al., 2024; OpenAI, 2024).

## 6  Conclusion and Limitation

We have presented STARFlow, the first latent based normalizing flow model that scales to high resolution images and large scale text to image modeling. Our results demonstrate that normalizing flows are scalable generative modeling method, and is capable of achieving comparable results to strong diffusion and autoregressive baselines.

There are also limitations to our work. For example, we have exclusively relied on pretrained autoencoders for simplicity, but it leaves the question of a potential joint latent–NF model design unexplored. Moreover, in this work we have primarily focused on training high-quality models, which comes at the cost of un-optimized inference speed. Additionally, our evaluation has been restricted to class- and text-conditional image generation on standard benchmarks; how well the approach generalizes to other modalities (e.g., video, 3D scenes) or more diverse, real-world data distributions remains to be seen.

## Acknowledgements

We thank Ying Shen, Yizhe Zhang, Navdeep Jaitly, Alaa El-Nouby and Preetum Nakkiran for helpful discussions. We also thank Samy Bengio for leadership support that made this work possible.

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

# Appendix

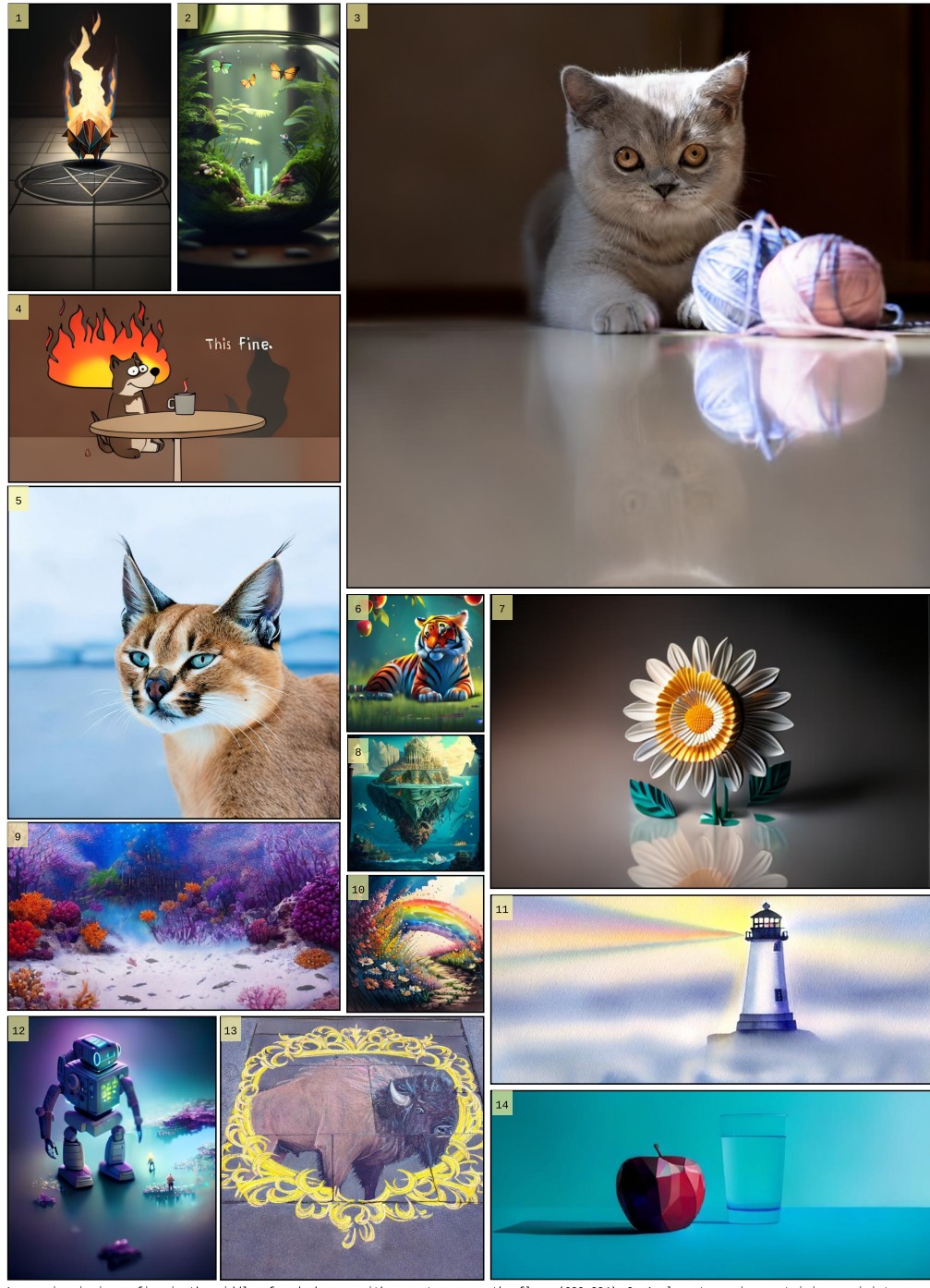

1.an origami pig on fire in the middle of a dark room with a pentagram on the floor (680x384); 2. A glass terrarium containing a miniature rainforest ecosystem, complete with tiny waterfalls, exotic plants, small animals like frogs and butterflies, the glass reflecting light from a nearby window. (680x384); 3. A British shorthair kitten playing with yarn in a room batched in sunlight (1024x1024); 4. smiling cartoon dog sits at a table, coffee mug on hand, as a room goes up in flames. "This is fine," the dog assures himself. (384x680); 5. Photorealistic close-up of a caracal gliding in the icy Antarctic shelf (1024x1024); 6. a beautiful tiger pokemon under an apple tree, cartoon style (256x256); 7. A daisy flower made entirely of origami paper, placed against a minimalist background, showcasing the folds and craftsmanship, high-resolution, studio lighting. (416x624); 8. A detailed painting of Atlantis, featuring intricate detailing and vibrant colors (256x256); 9. A vaporous coral metropolis embedded in frozen time, vivid watercolor bloom (384x680); 10. A watercolor painting of a vibrant flower field in spring, with a rainbow of blossoms. (256x256); 11. A lighthouse emitting rainbow beams into coastal fog, watercolor illustration, bathed in golden hour light (336x784); 12. Soft pastel painting of a robotics engineer by a bioluminescent tide pool, top-to-bottom visual flow, tilt-shift miniaturization effect, photoreal 8K detail (576x456); 13. chalk pastel sidewalk mural of a bison in ornate golden frame (512x512); 14. A red apple on a blue table next to a glass of water, low-poly 3-D art (336x784).

Figure 10: Selected samples of various aspect ratios from STARFlow on for text-to-image generation ($\omega = 4.0$). Image resolutions are adjusted proportionally for the ease of visualization.

# A  Derivations

## A.1  Extended Discussion of Prop. 1

**Why a Single Block ($T = 1$) Cannot Be Universal.**  With only one autoregressive–flow block,

$$\boldsymbol{x}_d \;=\; \mu_\theta(\boldsymbol{x}_{<d}) \;+\; \sigma_\theta(\boldsymbol{x}_{<d})\,\boldsymbol{z}_d, \qquad \boldsymbol{z}_d \sim \mathcal{N}(0,1),\; d = 1,\ldots,D,$$

each conditional $p(\boldsymbol{x}_d \mid \boldsymbol{x}_{<d})$ is *necessarily a single Gaussian*. Because no latent variable influences the affine parameters *beyond* the current coordinate, the model cannot represent multimodal densities or heavy tails. Consequently, $T = 1$ flows are *not* dense in $L^1(\mathbb{R}^D)$ and fail the universal-approximation criterion.

**Why $T = 2$ Is Sufficient.**  For $T = 2$ blocks with *opposite* orderings, all coordinates except the last ($d < D$) are expressed as infinite Gaussian mixtures (Eq. (5)) and hence enjoy the universal-approximation property via the density of Gaussian mixtures (Goodfellow et al., 2016). The principal reason why $\boldsymbol{x}_D$ fails to possess the universal approximation property lies in the structure:

$$\boldsymbol{x}_D = \mu_\theta^b(\boldsymbol{x}_{<D}) + \sigma_\theta^b(\boldsymbol{x}_{<D}) \cdot \boldsymbol{y}_D, \tag{10}$$

where $\boldsymbol{y}_D$ is defined as

$$\boldsymbol{y}_D = \mu_\theta^a(\boldsymbol{y}_{>D}) + \sigma_\theta^a(\boldsymbol{y}_{>D}) \cdot \boldsymbol{z}_D \tag{11}$$

$$= \mu_\theta^a(\varnothing) + \sigma_\theta^a(\varnothing) \cdot \boldsymbol{z}_D. \tag{12}$$

It is evident that $\boldsymbol{y}_D$ follows a unimodal Gaussian, since $\boldsymbol{z}_D$ is sampled from a unimodal Gaussian prior and the functions $\mu_\theta^a$ and $\sigma_\theta^a$ receive no random variable input, regardless of their nonlinearity. Consequently, $\boldsymbol{x}_D$ also becomes a unimodal Gaussian, inheriting this limitation from $\boldsymbol{y}_D$. Moreover, the above derivation only uses the *base* assumption $\boldsymbol{z}_d \sim \mathcal{N}(0,1)$ in the generative direction. In general, conditioning on observed coordinates induces a non-Gaussian latent distribution (i.e., $q_\theta(\boldsymbol{z}_d \mid \boldsymbol{x}_{<d})$ is not necessarily Gaussian). Consequently, the resulting conditional $q_\theta(\boldsymbol{x}_D \mid \boldsymbol{x}_{<D})$ can be even more complex than a single Gaussian.

In summary, $T = 1$ flows are fundamentally limited to unimodal Gaussians; $T = 2$ flows with alternating orderings achieve universality on $D - 1$ coordinates but leave the final one unimodal; and $T \geq 3$ flows overcome this last obstacle, granting full universal approximation power.

## A.2  Proof of Prop. 2

*Proof.* For an isotropic Gaussian $p(\boldsymbol{x}) = \mathcal{N}(\mu, \sigma^2 I)$ the score is

$$\nabla_{\boldsymbol{x}} \log p(\boldsymbol{x}) \;=\; -\frac{\boldsymbol{x} - \mu}{\sigma^2}.$$

Hence

$$\nabla_{\boldsymbol{x}} \log p_c(\boldsymbol{x}) = -\frac{\boldsymbol{x} - \mu_c}{\sigma_c^2}, \qquad \nabla_{\boldsymbol{x}} \log p_u(\boldsymbol{x}) = -\frac{\boldsymbol{x} - \mu_u}{\sigma_u^2}.$$

**Step 1: Guided score.** Insert these into Eq. (8) (CFG):

$$\nabla_{\boldsymbol{x}} \log \tilde{p}_c(\boldsymbol{x}) = (1 + \omega)\left(-\frac{\boldsymbol{x} - \mu_c}{\sigma_c^2}\right) \;+\; \omega\left(\frac{\boldsymbol{x} - \mu_u}{\sigma_u^2}\right)$$

$$= -\left[\left(\tfrac{1+\omega}{\sigma_c^2} - \tfrac{\omega}{\sigma_u^2}\right)\boldsymbol{x} \;-\; \left(\tfrac{1+\omega}{\sigma_c^2}\mu_c - \tfrac{\omega}{\sigma_u^2}\mu_u\right)\right]. \tag{13}$$

**Step 2: Match to a Gaussian form.** Any Gaussian $\mathcal{N}(\tilde{\mu}_c, \tilde{\sigma}_c^2 I)$ has score $-(\boldsymbol{x} - \tilde{\mu}_c)/\tilde{\sigma}_c^2$. Equating with Eq. (13) gives, for all $\boldsymbol{x}$,

$$\frac{1}{\tilde{\sigma}_c^2} = \frac{1 + \omega}{\sigma_c^2} - \frac{\omega}{\sigma_u^2}, \tag{14}$$

$$\frac{\tilde{\mu}_c}{\tilde{\sigma}_c^2} = \frac{1 + \omega}{\sigma_c^2}\mu_c - \frac{\omega}{\sigma_u^2}\mu_u. \tag{15}$$

**Step 3: Solve for $\tilde{\sigma}_c$.** Let $s := \sigma_c^2 / \sigma_u^2 (> 0)$. Rewrite Eq. (14):

$$\frac{1}{\tilde{\sigma}_c^2} = \frac{(1 + \omega) - \omega s}{s\,\sigma_u^2} \implies \tilde{\sigma}_c^2 = \frac{s\,\sigma_u^2}{(1 + \omega) - \omega s} = \frac{\sigma_c^2}{1 + \omega - \omega s},$$

so that

$$\boxed{\tilde{\sigma}_c = \frac{\sigma_c}{\sqrt{1 + \omega - \omega s}}}.$$

**Step 4: Solve for $\tilde{\mu}_c$.** Multiplying Eq. (15) by $\tilde{\sigma}_c^2$ and substituting the expression above yields

$$\tilde{\mu}_c = \frac{(1 + \omega)\mu_c - \omega s\,\mu_u}{1 + \omega - \omega s} = \mu_c + \frac{\omega s}{1 + \omega - \omega s}\,(\mu_c - \mu_u).$$

$\square$

**Additional Discussion.**

- **Consistency with standard CFG.** When the two Gaussians share the same variance ($\sigma_c = \sigma_u \implies s = 1$), Eq. (9) reduces to $\tilde{\sigma}_c = \sigma_c$ and $\tilde{\mu}_c = \mu_c + \omega(\mu_c - \mu_u)$, exactly matching the conventional CFG used in diffusion models (Ho & Salimans, 2021).

- **Numerical stability.** The denominator $1 + \omega - \omega s$ can approach 0 or even become negative when $s$ is large, causing $\tilde{\sigma}_c^2$ to blow up or change sign. Intuitively, guidance should *sharpen* $p_c$, which entails $\tilde{\sigma}_c^2 \le \sigma_c^2$, i.e. $1 + \omega - \omega s \ge 1$. We therefore clip the variance ratio[3] to

$$s = \text{CLIP}(s, 0, 1),$$

guaranteeing $1 + \omega - \omega s \ge 1$ for any $\omega > 0$ and ensuring both numerical stability and a genuinely mode-seeking guided distribution.

## B  Implementation Details

### B.1  Architecture Design

**Overall Structure.**    We implement STARFlow with a decoder-only Transformer (Vaswani et al., 2017). The shorthand $l(N) - d$ (see § 3.2) denotes a single *deep* AF block of $l$ layers followed by $N - 1$ *shallow* blocks (two layers each) with hidden width $d$. Our class-conditioned baseline uses $18(6) - 2048$ ($\approx$1.4 B parameters), while the text-conditioned model uses $24(6) - 3072$ ($\approx$3.8 B parameters). Layer-allocation sweeps in Fig. 8(b–e) probe scalability and convergence. Unlike Zhai et al. (2024), we apply a final layer norm at the predictions of each Transformer block.

**Conditioning Mechanism.**    For both conditioning modes, the context is prepended as a prefix to the deep block, and we omit AdaLN (Peebles & Xie, 2023)—a choice that simplifies the network and marginally improves quality. ImageNet classes are provided as one-hot vectors. Text captions (T2I) are encoded by a frozen FLAN-T5-XL encoder (Raffel et al., 2020), truncated to 128 tokens.

**VAE Latent Space.**    Images are first mapped to continuous latent tokens via the DiT VAE (Peebles & Xie, 2023), which compresses spatial dimensions by $48\times$. Because performance is highly sensitive to patch size, we keep $p = 1$ for all resolutions, yielding sequences of 1024, 4096, and 16384 tokens for $256 \times 256$, $512 \times 512$, and $1024 \times 1024$ images, respectively. We also applied our proposed deep-shallow architecture in pixels (see Table 3). To match similar computation, we adopted a patch size of $p = 8$ for learning $256 \times 256$ images.

**Positional Embeddings.**    All variants employ rotary positional embeddings (RoPE) (Su et al., 2024); we adopt **3D-RoPE**, giving each token $(x, y, t)$, where $(x, y)$ encodes its spatial grid location ($(0, 0)$ for text tokens) and $t$ its caption index (0 for image tokens). During fine-tuning from $256 \times 256$ to higher resolutions, we align positions by setting $(x', y', t) = (x/\alpha, y/\alpha, t)$, where $\alpha$ is the up-sampling ratio.

---

[3]This is equivalent to clip the unconditional variance $\sigma_u$ when it is smaller than $\sigma_c$.

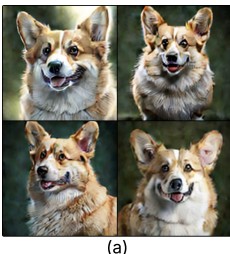 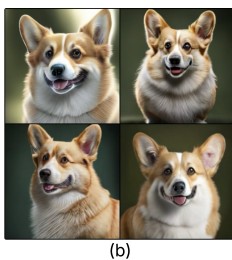 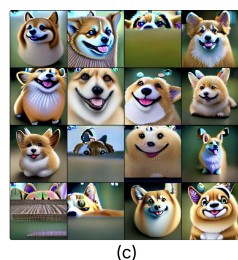 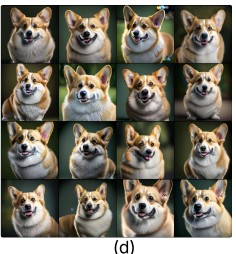

|(a)|(b)|(c)|(d)|

Figure 11: (a) Direct generation results using the model's latent samples without decoder fine-tuning or score-based denoising. (b) Results after applying decoder fine-tuning, effectively reducing latent-space noise. (c) and (d) provide comparisons of classifier-free guidance (CFG) strategies for text-to-image generation: (c) demonstrates degraded outputs at guidance weight $\omega = 5$ using the approach of Zhai et al. (2024), whereas (d) shows stable results with our proposed CFG method, confirming its improved effectiveness and suitability for text-conditioned applications.

**Default Configuration.**    Below is the default configurations of STARFlows:

```
model config for \model{}-l(N)-d:
    patch_size=1
    hidden_size=d
    num_layers=[l] + [2] * (N-1)
    num_channels_per_head=64
    use_swiglu_ffn=False
    use_rope=True
    use_final_rmsnorm=True
```

## B.2   Training Details

In all the experiments, we share the following training configuration for our proposed STARFlow. Models are trained on 32 (for 1.4B model) or 64 (for 3.8B model) H100 GPUs for around 2 weeks.

```
training config:
    batch_size=512
    optimizer='AdamW'
    adam_beta1=0.9
    adam_beta2=0.95
    adam_eps=1e-8
    learning_rate=1e-4
    min_learning_rate=1e-6
    learning_rate_schedule=cosine
    weight_decay=1e-4
    max_training_images=400M
    mixed_precision_training=bf16
```

**Stability of Eq. (3).**    The maximization term $-\log \sigma$ in Eq. (3) is *unbounded*: the model can drive some $\sigma$ values arbitrarily close to zero whenever this hardly influences $z$, echoing a classic pathology of normalizing-flow training. We mitigate it with three safeguards:

1. **Soft clipping.** Each raw Transformer output $\mathbf{x}$ is mapped through $f(\mathbf{x}) = a \tanh(\mathbf{x}/a)$, softly limiting its magnitude to $\pm a$.

2. **Positive scale parameterization.** The scale is enforced positive via $\sigma = \mathrm{softplus}(\hat{\sigma})$, where $\hat{\sigma}$ is the network's variance output.

3. **Latent norm penalty.** We add a small norm penalty over the intermediate latents $\boldsymbol{x}^t$ to avoid extremely large values. Typically a weight of 1e-4 is enough to keep the magnitude stable without hurting the performance.

Table 4: GenEval comparison across different methods.

| Method | Overall | Single Obj. | Two Obj. | Counting | Colors | Position | Color Attri. |
|---|---|---|---|---|---|---|---|
| **Diffusion Models** | | | | | | | |
| SDv1.5 (Rombach et al., 2022) | 0.43 | 0.97 | 0.38 | 0.35 | 0.76 | 0.04 | 0.06 |
| PixArt-$\alpha$ (Chen et al., 2023) | 0.48 | 0.98 | 0.50 | 0.44 | 0.81 | 0.08 | 0.07 |
| SDv2.1 (Rombach et al., 2022) | 0.50 | 0.98 | 0.51 | 0.44 | 0.85 | 0.07 | 0.17 |
| DALL-E 2 (Ramesh et al., 2022) | 0.52 | 0.94 | 0.66 | 0.49 | 0.77 | 0.10 | 0.19 |
| SDXL (Podell et al., 2023) | 0.55 | 0.98 | 0.74 | 0.39 | 0.85 | 0.15 | 0.23 |
| DALL-E 3 (Betker et al., 2023) | 0.67 | 0.96 | 0.87 | 0.47 | 0.83 | 0.43 | 0.45 |
| SD3 (Esser et al., 2024) | 0.74 | 0.99 | 0.94 | 0.72 | 0.89 | 0.33 | 0.60 |
| **Autoregressive Models** | | | | | | | |
| LlamaGen (Sun et al., 2024) | 0.32 | 0.71 | 0.34 | 0.21 | 0.58 | 0.07 | 0.04 |
| Chameleon (Team, 2024) | 0.39 | – | – | – | – | – | – |
| Show-o (Xie et al., 2024) | 0.53 | 0.95 | 0.52 | 0.49 | 0.82 | 0.11 | 0.28 |
| Emu3 (Wang et al., 2024) | 0.54 | 0.98 | 0.71 | 0.34 | 0.81 | 0.17 | 0.21 |
| **Normalizing Flows** | | | | | | | |
| STARFlow (Ours) | 0.56 | 0.97 | 0.58 | 0.47 | 0.77 | 0.20 | 0.34 |

**Mixed-Resolution Training.** During the high-resolution phase, STARFlow supports *mixed resolutions*, preserving each image's native aspect ratio. Because the backbone is a Transformer, variable sequence lengths are handled naturally, so no aggressive cropping is required; this better retains scene content and improves caption–image alignment. We bucket images into nine aspect-ratio bins: 21:9, 16:9, 3:2, 5:4, 1:1, 4:5, 2:3, 9:16, and 9:21 with the ratio appened in the caption:

```
{original_caption}\n in a {aspect_ratio} aspect ratio.
```

Image is center-cropped and resized so that its token count roughly matches that of a square reference. For a $512 \times 512$ target, we enforce $H \times W \approx 512^2$. This procedure stabilizes optimization, maximizes GPU utilization, and is used in conjunction with the 3D-RoPE alignment described above.

### B.3 Decoder Fintuning Details

We perform decoder fine-tuning by freezing the encoder and introducing controlled noise into the latent representations. The decoder is then trained using a standard autoencoder loss comprising L2, perceptual, and GAN losses. Training is conducted on ImageNet images at a resolution of 256×256 for 200K updates with a batch size of 64, utilizing a single node with 8 GPUs. To monitor performance, we compute an rFID by randomly sampling 50K real images, adding Gaussian noise with a standard deviation of 0.3, and directly decoding these perturbed images to compare with real images. Our resulting rFID is approximately 2.73, which exceeds the best achievable gFID from STARFlow at 2.40. This suggests current STARFlow with this finetuned decoder might have reached a performance ceiling under the specified noise conditions, highlighting an avenue for future exploration. Notably, although trained only on ImageNet at $256 \times 256$ resolution, the fine-tuned decoder can seamlessly generalize to arbitrary resolutions, aspect ratios, and text-to-image domains. See visual comparison in Fig. 11 (a) and (b).

### B.4 Inference Details

**Notation.** Let $\{f_b\}_{b=1}^B$ denote the autoregressive flow blocks ordered from *deep* ($b{=}1$) to *shallow* ($b{=}B$). Deep blocks are text-conditioned; shallow blocks are unconditional. Each block predicts a Gaussian head for the next token. The learnable start token is denoted $\mathbf{s} \in \mathbb{R}^C$ (`[SOS]`). A pretrained VAE decoder $D$ maps the final latent to image space.

**Sampling procedure.**

**Closed-form classifier-free guidance for Gaussian heads (Prop. 2).** Let $p_u(x) {=} \mathcal{N}(\mu_u, \Sigma_u)$ and $p_c(x) {=} \mathcal{N}(\mu_c, \Sigma_c)$ be the unconditional and conditional heads. For scale $\gamma {\geq} 0$,

$$\Sigma_g^{-1} = (1 - \gamma)\Sigma_u^{-1} + \gamma\Sigma_c^{-1}, \qquad \mu_g = \Sigma_g\big[(1 - \gamma)\Sigma_u^{-1}\mu_u + \gamma\Sigma_c^{-1}\mu_c\big]. \tag{16}$$

If $\Sigma_u {=} \Sigma_c$, then $\mu_g = \mu_u + \gamma(\mu_c - \mu_u)$ and $\Sigma_g = \Sigma_u$. In practice, Eq. equation 16 is applied only in deep blocks; shallow blocks remain unconditional.

---

**Algorithm 1** Sampling from STARFlow Models

---

**Require:** Prompt $y$, guidance scale $\gamma \geq 0$, blocks $\{f_b\}$, block types $\tau_b \in \{\mathsf{deep}, \mathsf{shallow}\}$, per-block length $T$
 1: Draw top latent seed $u^{(B+1)} \sim \mathcal{N}(0, I)$
 2: **for** $b = 1$ **to** $B$ **do**                                   ▷ deep block → shallow blocks
 3:      Initialize per-head KV caches $(\mathcal{K}_b, \mathcal{V}_b)$
 4:      **if** $\tau_b = \mathsf{deep}$ **then**
 5:          Prefill $(\mathcal{K}_b, \mathcal{V}_b)$ with a single forward pass over text embeddings $e(y)$
 6:      **else**
 7:          Zero-initialize $(\mathcal{K}_b, \mathcal{V}_b)$
 8:      **end if**
 9:      $x_1 \leftarrow \mathbf{s}$                            ▷ inject [SOS] at the first position in *every* block
10:      **for** $t = 1$ **to** $T$ **do**
11:          $(\mu_u, \Sigma_u) \leftarrow f_b(x_{1:t}; \mathcal{K}_b, \mathcal{V}_b, \mathrm{COND} = \varnothing)$
12:          **if** $\tau_b = \mathsf{deep}$ **and** CFG on **then**
13:              $(\mu_c, \Sigma_c) \leftarrow f_b(x_{1:t}; \mathcal{K}_b, \mathcal{V}_b, \mathrm{COND} = y)$
14:              $(\mu_g, \Sigma_g) \leftarrow \mathrm{GAUSSIANCFG}(\mu_u, \Sigma_u, \mu_c, \Sigma_c, \gamma)$
15:          **else**
16:              $(\mu_g, \Sigma_g) \leftarrow (\mu_u, \Sigma_u)$
17:          **end if**
18:          $x_{t+1} \sim \mathcal{N}(\mu_g, \Sigma_g)$; append keys/vals of $x_{t+1}$ into $(\mathcal{K}_b, \mathcal{V}_b)$
19:      **end for**
20:      $u^{(B-b+1)} \leftarrow \mathrm{REVERSE}(x_{1:T})$              ▷ hand off to the next block in reverse order
21: **end for**
22: **return** $D(\mathrm{RESHAPETO2D}(u^{(1)}))$

---

**Implementation details.** (i) [SOS] is used as the first input in *every* AR block at train and test time. (ii) KV caching follows standard LLM practice and is used only at inference in our code. (iii) Deep blocks maintain longer caches due to prompt prefill; shallow caches grow only with generated tokens.

**Optional parallel refinement.** For very long sequences, a Jacobi iteration style $K$-sweep variant can replace the inner loop by repeatedly updating all positions from a stale context and refreshing caches between sweeps. While faster wall-clock on some regimes, standard left-to-right sampling was most stable in our experiments.

### B.5    Baseline Details

**Diffusion Model Baseline** We deploy the official implementation of DiT[4] and report the performance. To make the architecture comparable to STARFlow, we set the number of layers to 28 and hidden dimension to 2048 while keeping the number of attention heads to 16, resulting in a model size of 2.1B parameters. We kept all of the other official repository settings the same. Notably the pretrained VAE of the official repository matches the one used in STARFlow. The baseline DiT is trained for 200M samples with batch size 256 using the official implementation settings: AdamW optimizer with learning rate 0.0001 and no weight decay 0.0. In inference, we set the number of sampling steps to 250 and classifier-free guidance scale to 1.5 following the best reported setting in the original paper.

**Autoregressive Model Baseline** We deploy the official implementation of LlamaGen[5] (Sun et al., 2024) and report the performance. In particular, to make the architecture comparable to our STARFlow, we set the number of layers as 28, hidden dimension 2048, and number of attention heads 32, which leads to the total model size of 1.4B parameters. We also adopt the VQ-VAE from the official repository with downsample factor 8 which matches the downsample factor used in STARFlow. The baseline LlamaGen is trained for 200M samples with batch size 512 using AdamW optimizer with learning rate 0.0001, weight decay 0.05 and betas $(0.9, 0.95)$. In inference, we set the

---

[4] https://github.com/facebookresearch/DiT
[5] https://github.com/FoundationVision/LlamaGen

top-k the same as the vocabulary size 16384 and temperature 1.0. We also implement classifier-free guidance with scale 1.75 following the best reported setting in the original paper.

## C   Additional Experiments

### C.1   Additional Evaluation on Text-to-Image Generation

Table 4 summarizes our GenEval (Ghosh et al., 2023) performance against representative diffusion and autoregressive (AR) baselines. STARFlow attains an Overall score of **0.56**—slightly above SDXL (0.55) and well ahead of earlier Stable Diffusion checkpoints—while simultaneously surpassing the several recent AR models for text-to-image generation, including Emu-3 (0.54), Chameleon (0.39), and LlamaGen (0.32). Improvements are most pronounced on the more compositional sub-tests. Crucially, these gains are achieved *without* any reward-based alignment, target-dataset finetuning, or caption rewriting—STARFlow is trained once, end-to-end, and evaluated exactly as generated. Because GenEval isolates visual grounding, we purposefully restrict comparison to image-only generators; nonetheless, STARFlow's single-pass inference already delivers substantial latency advantages over diffusion models that require tens to hundreds of denoising steps. The availability of exact log-likelihoods further opens avenues for principled preference learning, sequential planning, or cascaded generation—capabilities that likelihood-free baselines lack. An exciting next step is to couple STARFlow with large pretrained language- or vision–language models, forming a unified system that reasons jointly over text and images while retaining the speed, stability, and strong grounding demonstrated here.

### C.2   Inference Speed

Because STARFlow is autoregressive, tokens must be generated sequentially through every AF block, which makes inference latency the dominant bottleneck. Our deep–shallow redesign **partially** mitigates this issue: by concentrating parameters in the first few "deep" blocks and leaving the remaining ones lightweight with no condition or guidance, the incremental cost of later blocks becomes minimal. In practice, while the sampling speed is still relatively slow, this layout also outperforms the equal-sized architecture of Zhai et al. (2024) (see Table 5), and its overall runtime approaches that of a standard LLM—leaving additional head-room for techniques such as distillation or speculative decoding.

### C.3   Latent Denoising

A second limitation is that STARFlow cannot be trained directly on clean latents; adding Gaussian noise is required to keep the flow learning stable (Ho et al., 2019; Zhai et al., 2024), but this both complicates optimization and necessitates an explicit denoising stage at inference time. In this work, we investigated three strategies of denoising:

(1) **Single-step score denoising**. Use the flow itself as a score estimator and apply one denoising step. Works only for mild noise; at $\sigma = 0.3$ outputs are noticeably blurry.
(2) **Multi-step diffusion denoising**. Start from the noisy latent and run unconditional DDIM steps with a pretrained DiT. Quality improves, but latency and model complexity increase substantially.
(3) **Decoder finetuning (ours)**. Finetune the VAE decoder so it can reconstruct directly from noisy latents. Training can be done very efficiently on unconditional images, and the GAN objective effectively handles the uncertainty. This option is the simplest to deploy.

Future work will aim for a principled solution that trains directly on clean data, eliminating the denoising stage entirely.

Table 5: Per-block inference time (s) with a fixed batch size 16 of the 1.4B sized model for generating $256 \times 256$ images. Sampling speed is measured with CFG. The proposed deep–shallow uses **6** blocks: an 18-layer Transformer followed by a 5 blocks of 2-layer Transformer. The hidden dimension is 2048. The equal-sized (Zhai et al., 2024) baseline uses **8** blocks where each block has 8 layer of Transformers. To match the overall parameters, we reduce the hidden dimensions to 1280.

| Block ID | 0 | 1 | 2 | 3 | 4 | 5 | 6 | 7 | Total (s) |
|---|---|---|---|---|---|---|---|---|---|
| Equal-sized (Zhai et al., 2024) | 9 | 9 | 9 | 9 | 9 | 9 | 9 | 9 | 72 |
| Deep–shallow (ours) | 18 | 2 | 2 | 2 | 2 | 2 | - | - | 35 |

Table 6: Comparison of latent-denoising strategies at $\sigma_L = 0.3$ on ImageNet $256 \times 256$.

| Method | Extra Model | Extra Steps | FID 50K $\downarrow$ | Remarks |
|---|---|---|---|---|
| Single-step score | – | 1 | 2.96 | Blurry |
| Multi-step DiT (from 0.3) | DiT-XL | 30 | 2.53 | Slowest |
| Decoder finetune | Finetuned Decoder | 0 | **2.40** | Best, simplest |

# D  Application Details

## D.1  Training-free Image Inpainting with STARFlow

Let $M \in \{0, 1\}^{H \times W}$ be a binary mask that selects the pixels to be filled and let $\boldsymbol{x}_{\mathrm{gt}} \in \mathbb{R}^{H \times W \times C}$ be the ground-truth image (available only at evaluation time for measuring fidelity). We split the image into the *observed* part $\boldsymbol{x}_O = (1 - M) \odot \boldsymbol{x}_{\mathrm{gt}}$ and the *missing* part $\boldsymbol{x}_M = M \odot \boldsymbol{x}_{\mathrm{gt}}$. The pretrained flow $\boldsymbol{f}_\theta : \boldsymbol{x} \mapsto \boldsymbol{z}$ induces a tractable density $p_\theta(\boldsymbol{x}) = \mathcal{N}(\boldsymbol{f}_\theta(\boldsymbol{x}); \boldsymbol{0}, I) |\det \nabla_{\boldsymbol{x}} \boldsymbol{f}_\theta|$. To sample from the conditional $p_\theta(\boldsymbol{x}_M \mid \boldsymbol{x}_O)$ *without retraining*, we construct a Metropolis–Hastings (MH) chain in latent space:

1. **Init.** Replace the missing region by Gaussian noise, $\tilde{\boldsymbol{x}}^{(0)} = \boldsymbol{x}_O + M \odot \boldsymbol{\epsilon}$, $\boldsymbol{\epsilon} \sim \mathcal{N}(0, \sigma^2 I)$, and map to latent space $\boldsymbol{z}^{(0)} = \boldsymbol{f}_\theta(\tilde{\boldsymbol{x}}^{(0)})$.

2. **Proposal.** Draw fresh noise in the *same* masked region of latent space

$$\boldsymbol{z}' = \boldsymbol{z}^{(t)} + M \odot \boldsymbol{\gamma}, \qquad \boldsymbol{\gamma} \sim \mathcal{N}(0, \tau^2 I),$$

   and obtain the candidate image $\boldsymbol{x}' = \boldsymbol{f}_\theta^{-1}(\boldsymbol{z}')$. We then restore the context pixels, $\tilde{\boldsymbol{x}}' = \boldsymbol{x}_O + M \odot \boldsymbol{x}'$, ensuring every proposal satisfies the observed evidence.

3. **Acceptance.** Because the forward and reverse proposals are symmetric, the MH acceptance probability reduces to the ratio of conditional probabilities:

$$\alpha = \min\Big\{1, \frac{p_\theta(\tilde{\boldsymbol{x}}' \mid \boldsymbol{x}_O)}{p_\theta(\tilde{\boldsymbol{x}}^{(t)} \mid \boldsymbol{x}_O)}\Big\} = \min\Big\{1, \exp[\log p_\theta(\tilde{\boldsymbol{x}}') - \log p_\theta(\tilde{\boldsymbol{x}}^{(t)})]\Big\}.$$

   Accept with probability $\alpha$; otherwise keep the current state.

4. **Iteration.** Set $t \leftarrow t + 1$ and repeat steps (ii)–(iii) until convergence; the final sample $\tilde{\boldsymbol{x}}^{(T)}$ is reported as the inpainted image.

Intuitively, each step perturbs only the masked latents, letting the powerful flow prior propose content that is *globally* coherent with the context while the MH test enforces exact consistency with the joint density. The chain is ergodic—Gaussian noise gives non-zero probability to every latent configuration—and its stationary distribution is precisely $p_\theta(\boldsymbol{x}_M | \boldsymbol{x}_O)$. In practice we set both $\sigma = 1$ and $\tau = 1$. Since our STARFlow is well-trained on large-scale text-to-image data with sufficient capacity, it yields high acceptance rates and we set the total iterations to **20**. No additional training, guidance network, or data-specific tuning is required. effective plug-in for image inpainting with pretrained autoregressive flows.

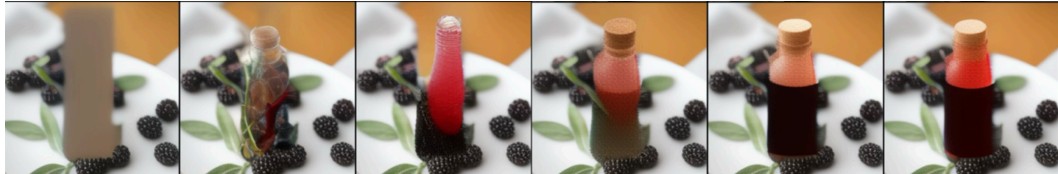

Figure 12: Demonstration of generation trajectories of inpainting output.

## D.2 Interactive Image Editing with STARFlow

STARFlow can be naturally extended to multi-round tasks such as interactive image editing. We start from a pretrained text-to-image checkpoint and finetune on the ANYEDIT corpus[6]. For simplicity, we use only the subset that provides text instructions. Each training quadruple $(\boldsymbol{x}^{\mathrm{src}}, \boldsymbol{t}^{\mathrm{cap}}, \boldsymbol{t}^{\mathrm{inst}}, \boldsymbol{x}^{\mathrm{tgt}})$ contains a source image, its caption, a free-form instruction, and the edited target (see Fig. 6b).

We serialize every sample into the sequence

$$\left[\mathrm{T5}(\boldsymbol{t}^{\mathrm{cap}}), \ \mathrm{AFs}\big[\mathrm{VAE}(\boldsymbol{x}^{\mathrm{src}})\big], \ \mathrm{T5}(\boldsymbol{t}^{\mathrm{inst}}), \ \mathrm{AFs}\big[\mathrm{VAE}(\boldsymbol{x}^{\mathrm{tgt}})\big]\right],$$

where image segments are tokenized by our VAE ($p = 1$) and text segments are embedded by a frozen FLAN-T5-XL (Raffel et al., 2020; Chung et al., 2022). Image latents first pass through the shallow-AF blocks independently, after which all tokens are processed by the shared deep-AF Transformer. Because the deep block is strictly causal, the edited image and all later tokens can attend to the entire prefix—including the source image—without any special masking. During inference the prefix is written once into the *KV* cache; sampling the edited tokens simply reads from this cache, mirroring the behavior of language-only LLMs.

**Joint Training Objective.** Instead of optimizing a single conditional likelihood, we maximize the *joint* log-likelihood of both images:

$$\max_{\theta} \mathcal{L}_{\mathrm{joint}} \ = \ \mathbb{E}_{(\boldsymbol{x}^{\mathrm{src}},\boldsymbol{t}^{\mathrm{cap}},\boldsymbol{t}^{\mathrm{inst}},\boldsymbol{x}^{\mathrm{tgt}})}\Big[\log p_{\theta}\big(\boldsymbol{x}^{\mathrm{src}} \mid \boldsymbol{t}^{\mathrm{cap}}\big) + \log p_{\theta}\big(\boldsymbol{x}^{\mathrm{tgt}} \mid \boldsymbol{t}^{\mathrm{inst}}, \boldsymbol{x}^{\mathrm{src}}, \boldsymbol{t}^{\mathrm{cap}}\big)\Big],$$

where each term is evaluated via the change-of-variables formula (Eq. (3)). This objective maintains maximum-likelihood training, allows gradients to propagate across *all* modalities, and enables the same network to generate from scratch (empty image prefix) or perform edits (given image prefix).

Unlike diffusion-based MLLMs that first *generate* pixels and then re-encode them with a separate vision backbone, our autoregressive flow is invertible: a single forward pass encodes the user image, and a single reverse pass decodes the edited result. Encoding and decoding share parameters, introduce no information loss, and integrate seamlessly with the Transformer's *KV* cache. This *single-pass round-trip* property sharply reduces latency and highlights autoregressive flows as a compelling choice for tightly coupled vision–language applications. We show interactive image generation and editing examples in Fig. 13 where given a caption and editing instruction, our model predicts two images one after another.

# E   Related Topic Discussion

## E.1   Autoregressive Model v.s. Autoregressive Flow

Connections between the two families emerge in masked autoregressive flows (MAF, (Papamakarios et al., 2017)), which impose invertibility on an autoregressive factorisation, yet fundamental differences remain. Autoregressive models dispense with any latent prior; each conditional distribution is learned directly in the data domain, which is typically discrete—tokens, integer pixels, or quantized audio samples—and generation proceeds strictly one element at a time. Normalizing flows, by contrast, begin from an explicit Gaussian prior in a continuous latent space and learn an invertible transformation that warps this prior into the target distribution. This design delivers exact log-likelihoods, parallel one-shot sampling, and bidirectional latent inference, but at the cost of enforcing invertibility and differentiability in every layer. While MAF narrows the gap by marrying an autoregressive factorisation with invertibility, the reliance on a Gaussian base and a continuous formulation remains the defining hallmark of normalizing flows, whereas the absence of a prior and the natural alignment with discrete data continue to characterise pure autoregressive models.

---

[6] https://dcd-anyedit.github.io

## E.2 Flow Matching v.s. Autoregressive Flow

Normalizing flows (NF) and Flow Matching (FM) both map a simple latent prior to the data distribution, but they differ fundamentally in what they optimise and how they realise the map. A normalizing flow learns a time-independent bijection whose parameters are updated by directly minimising the data's negative log-likelihood (NLL); the change-of-variables formula provides an exact, unbiased gradient, so every parameter update moves the model toward the true maximum-likelihood solution. Flow Matching instead specifies a time-dependent vector field that transports probability mass along a chosen path and trains this field with a velocity-matching loss. In short, Flow Matching reduces per-iteration cost by relaxing the objective, but Normalizing Flows retain the rigorous maximum-likelihood foundation, and exact densities.

## E.3 Relation to JetFormer (Tschannen et al., 2024b)

**Architectural differences.** JetFormer constructs a Transformer-RealNVP flow ("Jet") in *pixel* space and employs a GMM prior; training optimizes an ELBO due to the latent prior. In contrast, STARFlow applies uniform affine autoregressive flows throughout, is invertible end-to-end, and operates in the latent space of a fixed pretrained auto-encoder, so the flow itself is trained by exact likelihood while the overall objective can be viewed as a fixed-encoder ELBO.

**Expressivity and depth.** Prop. 1 establishes that a small number of autoregressive flow blocks already achieves universality in our construction (2–3 blocks suffice), which keeps STARFlow compact and scalable. In our experience, the Jet mapping typically requires many blocks for comparable expressivity.

**Priors and guidance.** A Gaussian prior in STARFlow enables the closed-form guidance in Eq. equation 16. While a GMM prior could be substituted, we found it unnecessary. Conversely, replacing JetFormer's GMM with a standard Gaussian weakens its modeling power under the Jet architecture (consistent with its reported ablations) and removes a straightforward path to closed-form guidance.

**Factor-out.** JetFormer adopts multi-scale factor-out to compress activations for its GMM prior. STARFlow does not factor-out by default; adopting a similar device is a natural extension but not required for our results.

Table 7: Concise comparison with JetFormer.

| Aspect | JetFormer | STARFlow |
|---|---|---|
| Domain | Pixels | Pretrained VAE latent |
| Transform | Transformer-RealNVP (partial invertibility) | Affine AR flows (fully invertible network) |
| Objective | ELBO (GMM prior) | Exact flow NLL (fixed-encoder ELBO overall) |
| Depth | Many Jet blocks often required | 2–3 AR blocks suffice (Prop. 1) |
| Prior | GMM | $\mathcal{N}(0, I)$ (enables Eq. 16) |
| Factor-out | Multi-scale factor-out | Not used by default |
| Guidance | No closed-form CFG | Closed-form Gaussian CFG |

# F Broader Impacts

**Positive societal impacts.** STARFlow shows—for the first time—that *normalizing flow*–based models can scale to the same resolutions and sample quality previously dominated by diffusion and discrete autoregressive methods. The invertibility of normalizing flows enables interactive image editing (See Fig. 9b for examples) making STARFlow suitable for assistive technologies (e.g., real-time diagram manipulation for education or accessibility) and for professional design workflows that demand faithful, iterative refinement.

**Potential risks and negative impacts.** Higher-quality image generation lowers the barrier to fabricating realistic—but false—visual evidence. Interactive editing magnifies this risk by enabling

rapid revision cycles. We advocate the concurrent development of reliable flow-specific watermarking and provenance tools.

## G  Additional Samples

We show more generated samples from STARFlow in Figs. 14 to 17.

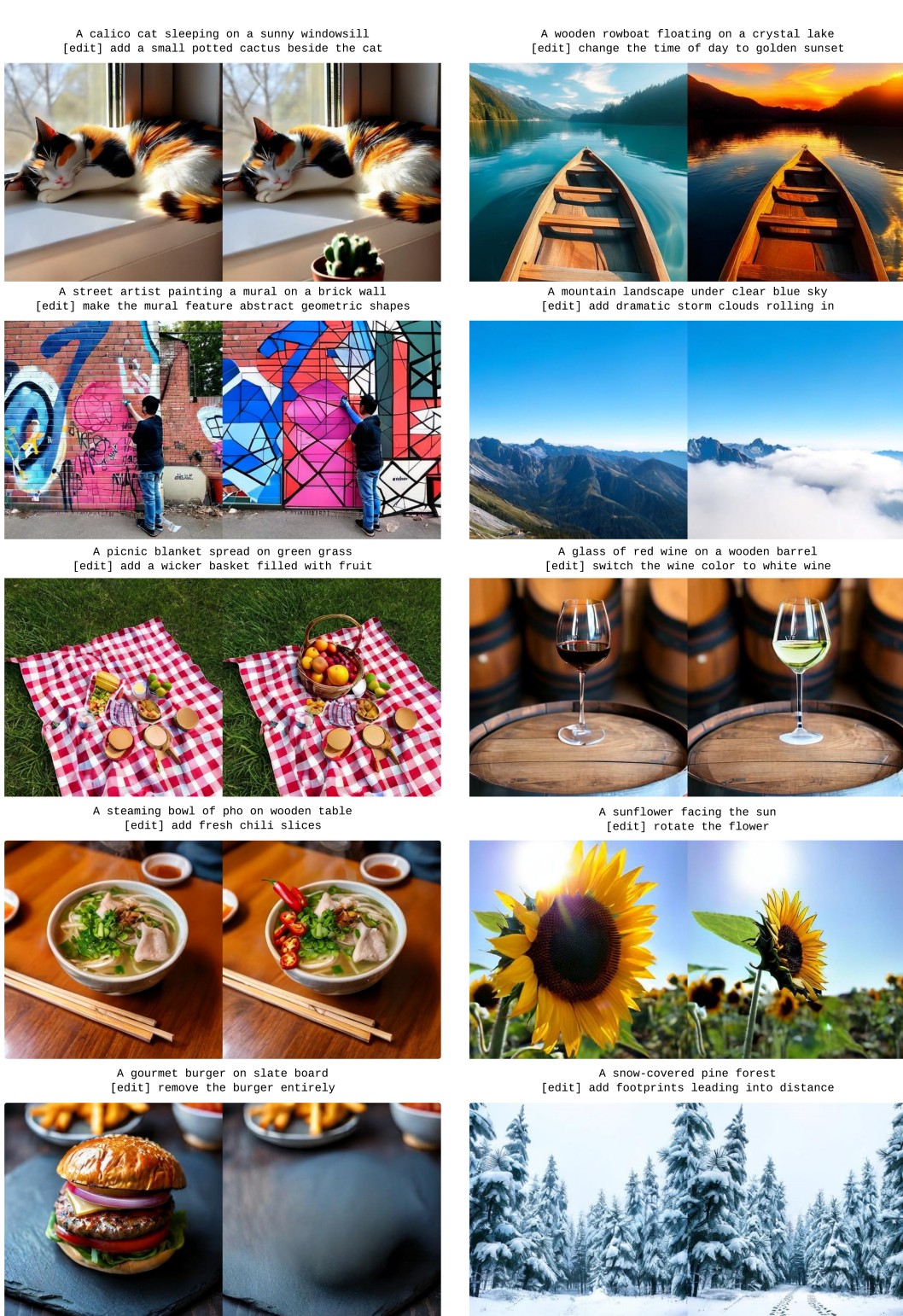

Figure 13: Interactive editing with STARFlow. Starting from an initial caption, STARFlow generates a base image. Given a subsequent user-provided editing instruction, the model then modifies the image accordingly—without requiring re-encoding. Each example illustrates a generic instruction applied to a generated image. All images are synthesized at a resolution of $512 \times 512$.

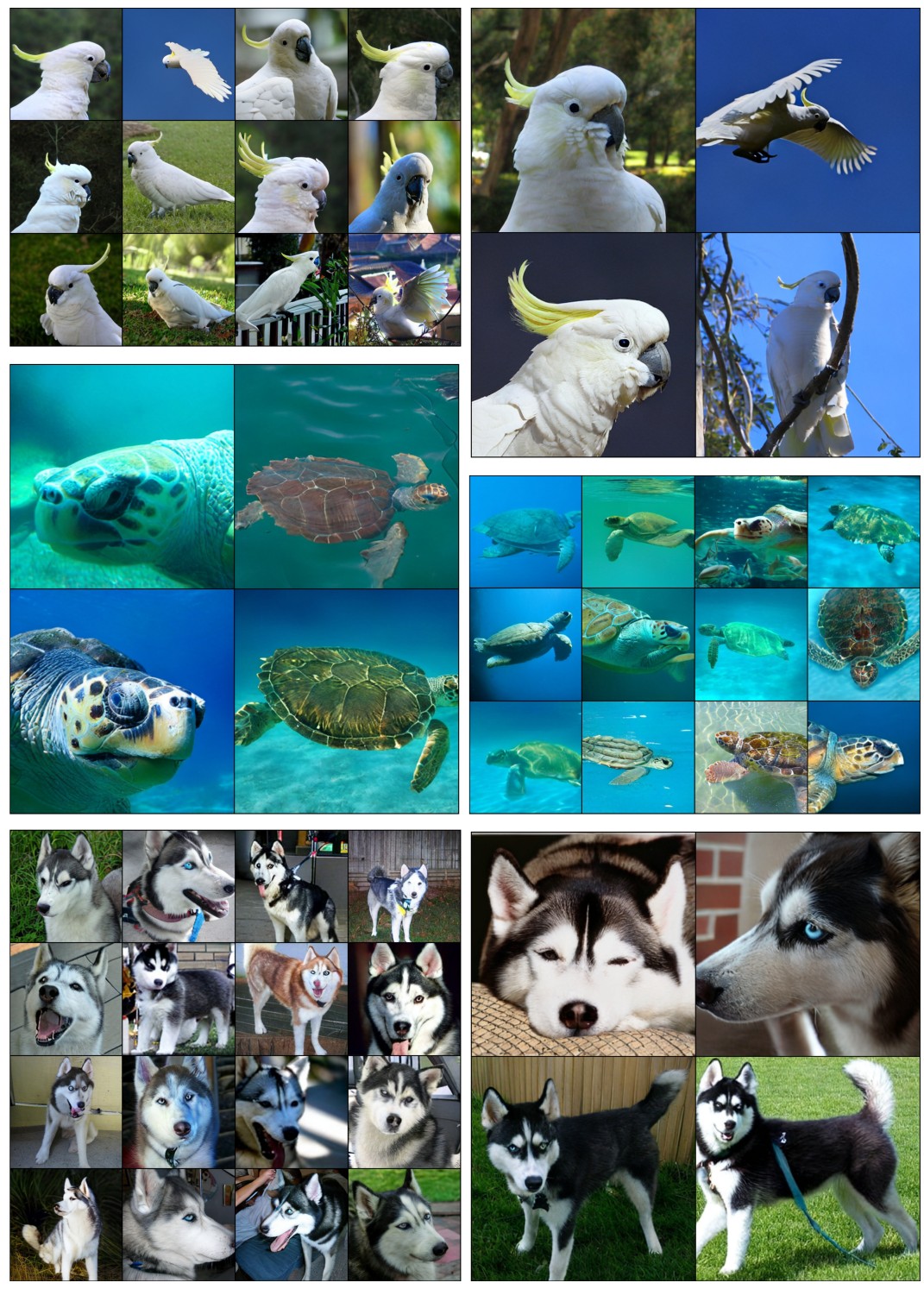

Figure 14: Additional class-conditioned generation from STARFlows trained on $256 \times 256$ and $512 \times 512$, respectively. The classes are *sulphur-crested cockatoo, Kakatoe galerita, Cacatua galerita*, *loggerhead*, *loggerhead turtle, Caretta caretta*, and *Siberian husky*.

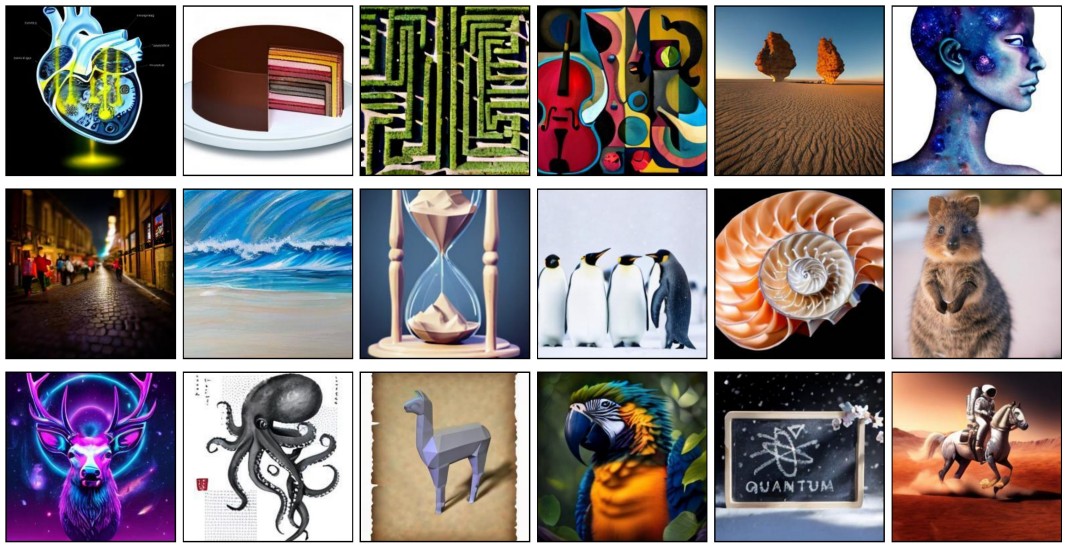

**1.** Schematic cutaway of a clockwork heart pumping luminous liquid, technical drawing style; **2.** Cross-section illustration of a layered cake that resembles planetary strata; **3.** Surreal desert with floating sandstone monoliths casting long shadows at golden hour, ultra-wide lens; **4.** Cubist still life of fruit and musical instruments, vivid complementary colors; **5.** Top-down shot of a labyrinth garden trimmed into Escher-like impossible geometry; **6.** Watercolor portrait of an abstract humanoid with translucent skin revealing galaxies; **7.** Tilt-shift photo of a festival lantern parade through narrow cobblestone streets; **8.** Oil-on-canvas seascape where waves are brush strokes of pure geometry; **9.** Design an hourglass where sand forms miniature mountains in low-poly 3-D model style; **10.** Juvenile emperor penguins huddling together on Antarctic ice shelf, gentle snowfall; **11.** Low-key studio shot of concentric nautilus shell cross-section revealing logarithmic spiral; **12.** Quokka standing on hind legs engaging camera with curious expression, beach background; **13.**Neon synthwave poster featuring a deer amid swirling galaxies; **14.** Japanese ink wash of an octopus surrounded by geometric patterns; **15.** Low-poly 3-D render of an alpaca on vintage parchment; **16.** Photorealistic close-up of a macaw hunting silently in the tropical rainforest canopy; **17.** High-resolution chalkboard typography sketch spelling 'Quantum' amid cherry-blossom snowfall, dramatic lighting; **18.** A photorealistic image of an astronaut riding a horse on Mars.

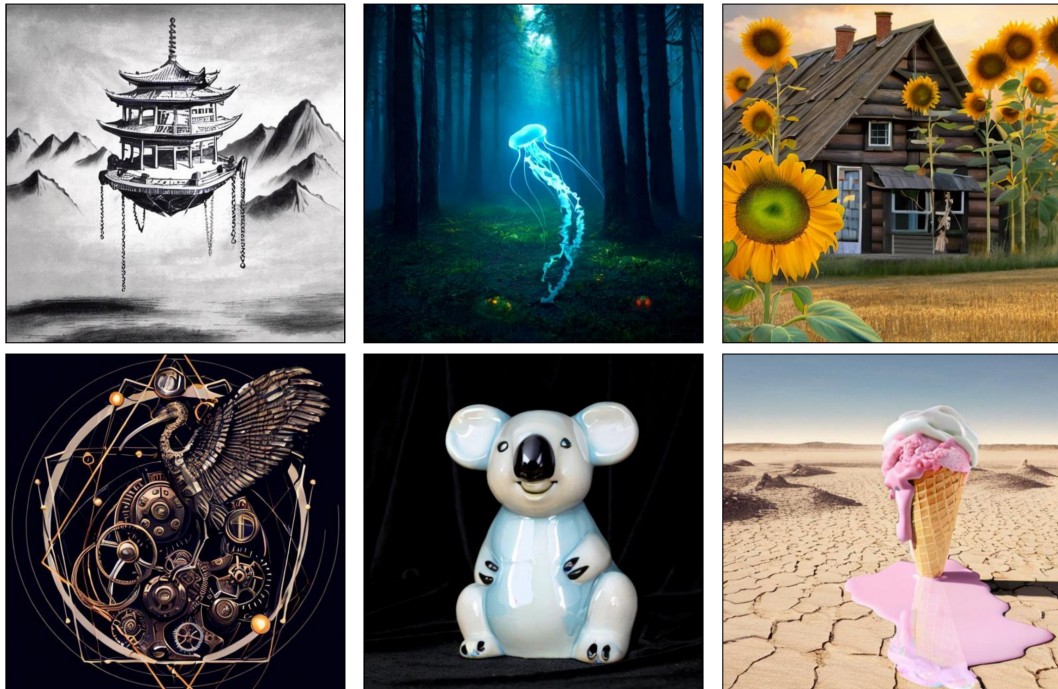

**1.** Ink-on-parchment concept art of a floating pagoda tethered by chains to mountain peaks; **2.** Glowing jellyfish drifting through a misty pine forest at dawn, photoreal composite; **3.** Timber-frame hobbit-style cottage under giant sunflowers, golden afternoon; **4.** Steampunk clockwork version of an ibis surrounded by geometric patterns; **5.** Ceramic glazed statue of a koala against black velvet backdrop; **6.** Dreamlike image of an ice cream cone melting into a desert landscape, surrealism.

Figure 15: Additional text-conditioned generation from STARFlows trained on $256 \times 256$ and $512 \times 512$, respectively.

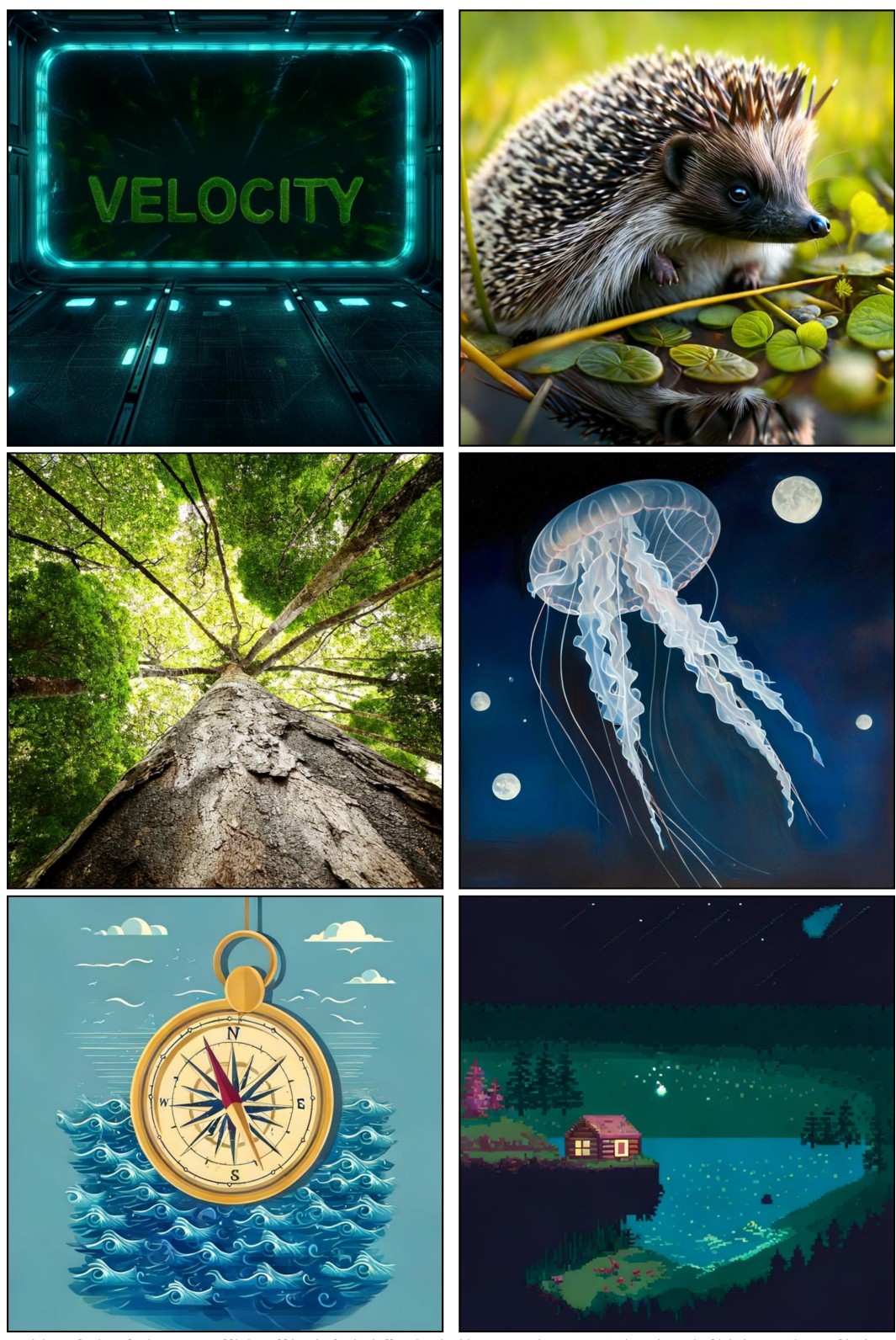

1. High-resolution glowing moss graffiti spelling 'Velocity' floating inside zero-gravity space station, dramatic lighting; 2. Photorealistic close-up of a hedgehog basking in sun in the lush river delta; 3. Ultra-wide rainforest canopy shot looking straight up at towering kapok trees and lianas; 4. baroque oil painting featuring a jellyfish under moonlit sky; 5. Design an antique compass floating above stormy seas in vector flat design style; 6. Retro 8-bit pixel art scene of a cozy lakeside cabin under meteor shower

Figure 16: Additional text-conditioned generation from STARFlows trained on $1024 \times 1024$.

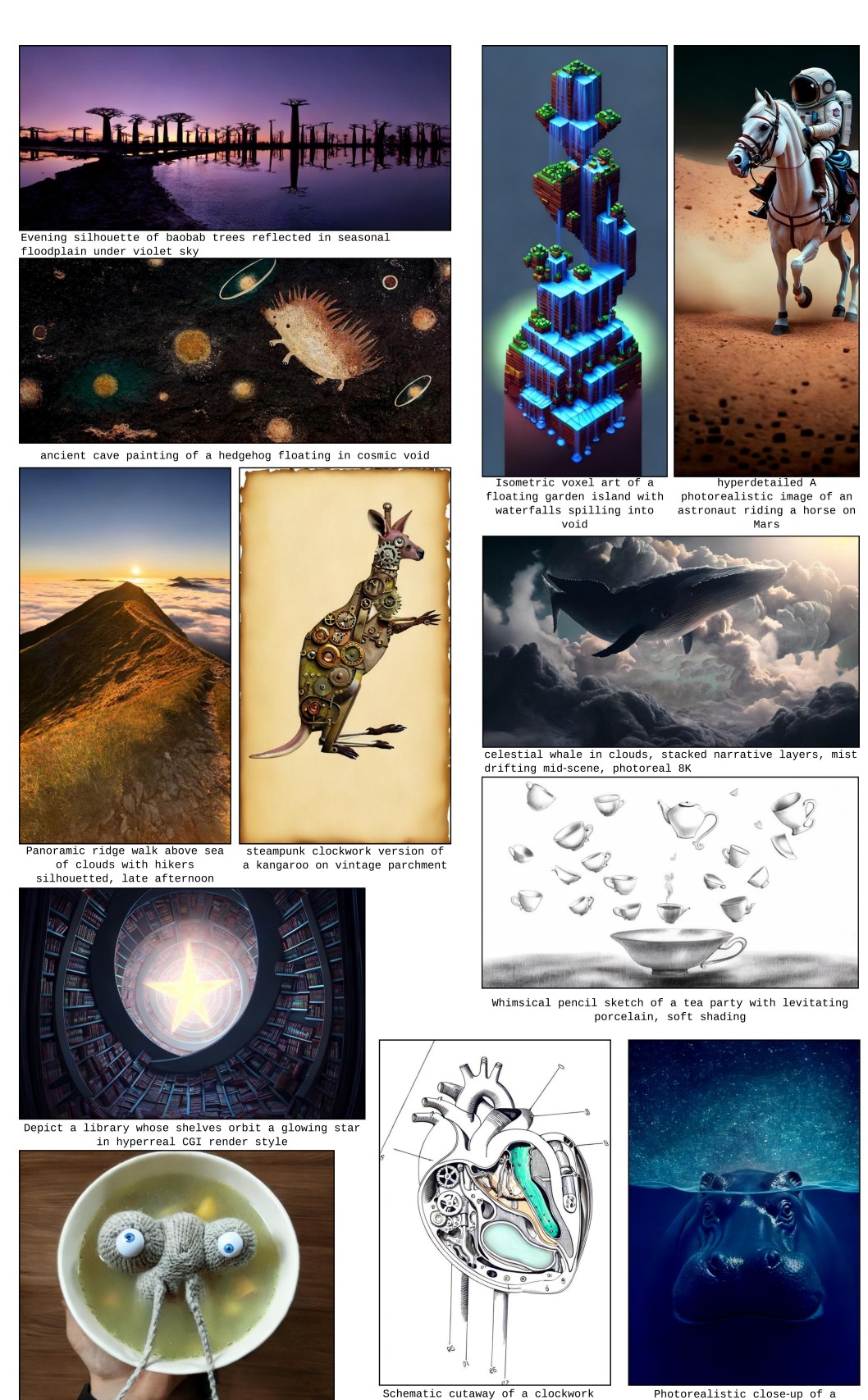

Figure 17: Additional text-conditioned samples from STARFlows trained on various aspect ratios.

