# OpenReview forum: "STARFlow: Scaling Latent Normalizing Flows for High-resolution Image Synthesis"
_NeurIPS.cc/2025/Conference — NeurIPS 2025 spotlight_

### Official Review · Reviewer_Zv2v · 2025-06-09

**Clarity:** 3
**Significance:** 3
**Originality:** 3
**Rating:** 5
**Confidence:** 3

**Summary:**

The authors introduce STARFlow, a scalable generative model grounded in autoregressive flows (AFs). They first prove the theoretical ability of AFs to model continuous distributions. Then, they propose several innovations to boost the scalability of normalizing flows: a deep-shallow architecture balancing capacity and efficiency, latent-space learning via pretrained autoencoders instead of direct pixel modeling, and a new guidance algorithm for better sample quality. Notably, STARFlow remains a single, end-to-end normalizing flow enabling exact continuous-space maximum likelihood training. The model shows competitive performance in class-to-image and text-to-image generation, with sample quality close to leading diffusion models. This work represents the first successful application of normalizing flows at such a large scale and high resolution.

**Questions:**

1. Could you elaborate on the inference process details with Classifier-Free Guidance (CFG)? Specifically, does Starflow apply CFG **only to each deep block**, or **to intra-layers within each block**?
2. STARFlow's SFT performance on GenEval and DPG benchmark. As Blip-3o open-sourced a high-quality SFT dataset, this would be a good SFT data choice[1].
3. Could you provide more visualizations or online demos?
4. Can the authors discuss the advantages compared to latent diffusion models?

[1] https://huggingface.co/datasets/BLIP3o/BLIP3o-60k

**Ethical Concerns:**

["NO or VERY MINOR ethics concerns only"]

**Final Justification:**

After carefully reviewing the author’s response, I have decided to maintain the current score (Accept).

**Limitations:**

see weakness

**Quality:**

3

**Strengths And Weaknesses:**

Strengths
1. The writing is clear and easy to understand.
2. The proposed deep-shallow structure sounds reasonable.
2. The pixel decoder seems natural and makes sense. Traditional Langevin sampling and diffusion methods involve heavy computational costs, which restrict their applications.
3. The performance improvement is significant.

Weaknesses
1. The performance of T2i on the GenEval benchmark appears inferior. I conjecture that a proper supervised fine-tuning (SFT) stage could significantly boost its performance.
2. The inference resources, including memory usage, time consumption, and GFlops, should be reported.
3. The scaling property of STARFlow versus parameters or Flops.

---

> ### Author Rebuttal · Authors · 2025-07-31
>
> We sincerely thank the reviewer for their constructive feedback and thoughtful evaluation. We are pleased that you found the writing clear, the deep–shallow design reasonable, and the performance improvements meaningful. Below, we address your comments in detail.
>
> ---
>
> # Weaknesses
>
> ## 1. GenEval Performance and Supervised Fine-Tuning (SFT)
>
> We thank the reviewer for this valuable suggestion regarding supervised fine-tuning (SFT). We would like to clarify that all current numerical results were obtained without any SFT or Reinforcement Learning (RL) steps, which relies solely on the pre-training data.
>
> We fully agree that applying SFT—particularly with powerful instruction datasets like those from recent work such as BLIP-3o [1]—would significantly improve performance. Thank you for highlighting this important direction. We will add a discussion to our paper and plan to include results from SFT and RL fine-tuning in future work to further improve our benchmark scores.
>
>
> ---
>
> ## 2. Reporting of Inference Resources
>
> We agree that a more detailed analysis of training and inference efficiency would strengthen the work. While we briefly discussed this in the limitations section, we will enhance the supplementary material with:
>
> - Inference time per image across STARFlow variants and all the baselines with a table comparison.
> - Training and sampling FLOPs per iteration
> - A runtime vs. sample quality trade-off analysis
>
>
> ---
>
> ## 3. Scaling Efficiency of STARFlow
>
> We acknowledge that understanding STARFlow’s performance scaling with model size and computational cost is important.
> We beliefly showed the scaling curves againt the model sizes in Fig 8.
> We have also conducted several ablation studies in this direction and will include these results in the revised version. Thank you for pointing this out!
>
> ---
>
> ## 4. Application of CFG Across STARFlow Blocks
>
> The classifier-free guidance (CFG) mechanism proposed in Proposition 2 is applied exclusively at the output of the deep blocks. Here is a more detailed breakdown:
>
> * **Conditioning and Purpose:** In our implementation, only the deep blocks accept conditioning inputs. The purpose of CFG is to integrate the **conditional prediction** (using the text prompt) and the **unconditional prediction** (using a null token) to produce an output that more strongly reflects the guidance.
> * **Mechanism:** At each autoregressive step within a deep block, the final layer predicts two sets of parameters: $(\mu_c, \sigma_c)$ for the conditional case and $(\mu_u, \sigma_u)$ for the unconditional case. The CFG function then recalculates these into a single, guided output: $\tilde{\mu}, \tilde{\sigma} = f_{CFG}(\mu_c, \sigma_c, \mu_u, \sigma_u)$.
> * **Derivation:** The specific function $f_{CFG}$ used in our paper, as shown in Proposition 2, is mathematically derived from the score function perspective of the deep block. We note that alternative implementations may also be possible.
> * **Application:** Analogous to guidance in diffusion models, the intermediate layers of a block operate independently of the CFG calculation. Guidance is applied only to the final output of a step. This guided output then determines the input for the subsequent autoregressive step via the equation $x = \tilde{\mu} + \tilde{\sigma} \cdot z$.
>
> ---
>
> ## 5. Additional Visualizations and Demos
>
> We will include more qualitative examples in the new appendix, including samples for GenEval-specific queries. Additionally, we are planning to release an online demo to better showcase STARFlow’s capabilities.
>
> ---
>
> ## 6. Comparison with Latent Diffusion Models
>
> Below, we summarize STARFlow’s advantages over latent diffusion models:
>
> - **Stronger likelihood guarantee** through direct maximum likelihood training in continuous latent space.
> - **Faster convergence** in terms of FID, using the same backbone (see Fig. 8a).
> - **End-to-end trainability**, which opens new avenues for research in image generation.
> - **Higher sampling throughput** due to autoregressive flow structure.
> - **Easier to combine with LLMs** due to the shared autoregressive structure in deep blocks.
>
> ---
>
> # Reference
>
> [1] [BLIP-3o Dataset on Hugging Face](https://huggingface.co/datasets/BLIP3o/BLIP3o-60k)
>
> ---
>
> We thank the reviewer again for the thoughtful suggestions. We believe the planned revisions and additional results will help address the concerns and further strengthen the paper.

---

### Official Review · Reviewer_382B · 2025-06-22

**Clarity:** 3
**Significance:** 4
**Originality:** 3
**Rating:** 5
**Confidence:** 4

**Summary:**

The authors show for the first time that autoregressive flow models can be scaled to deliver compelling high-resolution text-to-image generations approaching the level of quality of diffusion-based models. This result is achieved through a thorough theoretical and technical analysis of autoregressive flow models and underlying architecture designs. This analysis results in an improved deep-shallow design for the model backbone, a principled CFG guidance mechanism and the possibility to train the model in the latent space of an VAE. Experimental results show extensive ablations for all the proposed components and comparison on ImageNet 256, 512 and text-conditioned image generation, producing promising results. Finally, image inpainting and editing results are shown. The results presented in the work pave the way for a renewed exploration of normalizing flows as alternative generation frameworks to the current state-of-the-art.

**Questions:**

Main questions affecting my rating:
- Will the code be open sourced? While the authors answer [Yes] to the "Open access to data and code" section of the checklist, the justification suggests that code will not be released. The absence of an open source implementation might hamper further development of this research direction, reducing significance.
- A missing aspect in the paper is an analysis of training and inference costs with respect to baseline methods in Tab. 1, 2 and 3. This is not a critical issue, and the aspect is briefly touched upon in the limitations section. I suggest a thorough inclusion of training and inference flops and times as part of the experimental section to further strengthen the work.

Minor doubts not affecting my rating:
- The method requires application of noise to the latent space with a high sigma value of 0.3 (LL151 and LL153). This is fairly discussed as a limitation in the manuscript. The employed SD auto encoder is already trained as a VAE, making it possible to sample Z from the posterior distribution, resulting in a similar application of noise. Could the authors discuss why posterior sampling alone would not be a sufficient source of noise?
- Eq. (7) shows noise being applied in the pixel space, before the Encoder, while LL151 discusses application of the noise to the latent space. Could the authors clarify this aspect?
- LL208 why is the T5-XL encoder preferred over the XXL variant?

**Ethical Concerns:**

["NO or VERY MINOR ethics concerns only"]

**Final Justification:**

The author's provided answer on inclusion of performance metric and confirmation that the code will be open sourced make me keep my positive rating and I continue to argue for acceptance.

**Limitations:**

- Limitations are discussed in detail in the supplementary material

**Paper Formatting Concerns:**

- No typos were found in the paper and the paper is clearly written

**Quality:**

3

**Strengths And Weaknesses:**

QUALITY
- The paper takes a holistic view at autoregressive normalizing flows and thoroughly analyzes their design both theoretically and empirically. The analysis results in a model approaching performance of state-of-the-art diffusion-based generators.
- A missing aspect in the paper is an analysis of training and inference costs with respect to baseline methods in Tab. 1, 2 and 3. While inference times are informally discussed as a limitation in the supplementary material, a thorough benchmarking of training flops, inference flops and inference times for the proposed method and baselines reported in Tab. 1, 2 and 3 would further strengthen the experimental section.

CLARITY
- The paper is clearly explained and written. Figures summarize the framework intuitively, theoretical results are clearly presented and accompanied by proofs. The supplementary material discussed implementation details and limitations extensively.

SIGNIFICANCE
- The paper shows for the first time that normalizing flows can approach the quality of state-of-the-art methods in text-to-image image generation. While a quality gap still exists, the work is highly significant as it provides strong theoretical and empirical results that can pave the way for further development of normalizing flows as alternative generation frameworks to the current state-of-the-art.
- A series of theoretical results are proven that characterize the expressive power of normalizing flows and derive a principled CFG mechanism for them
- Extensive ablations are shown into model design that will be informative to guide future work in this research direction
- The feasibility of performing downstream applications such as image inpainting and editing is shown

ORIGINALITY
- The work is the first to scale autoregressive normalizing flows, reaching a level of quality approaching large-scale text-conditioned image generators.
- The authors refine autoregressive normalizing flows through their analysis. Novel contributions such as a the deep-shallow design and a principled CFG mechanism are introduced

---

> ### Author Rebuttal · Authors · 2025-07-31
>
> We sincerely thank the reviewer for their positive evaluation and constructive feedback. We are encouraged by your recognition of our contributions in advancing autoregressive normalizing flows, and we appreciate your comments on the theoretical, empirical, and architectural aspects of our work. Below, we address your specific comments in detail.
>
> ---
>
>
> ## 1. Request for Inference Time and Training FLOPs
>
> We agree that a more thorough quantitative analysis of training and inference efficiency would strengthen the paper. While we provided a brief discussion in the limitations section, we acknowledge the importance of presenting more concrete numbers. In the revision:
>
> - We will expand the supplementary material to include:
>   - Inference time per image for STARFlow and all baselines
>   - Training and sampling FLOPs per iteration
>   - A runtime vs. sample quality trade-off comparison across methods
>
> We believe these additions will provide a clearer understanding of STARFlow’s efficiency profile.
>
> ---
>
> ## 2. Code Availability
>
> **We will release the code on public**, and the checkpoint release is subject to legal related consideration.
>
> ---
>
> ## 3. Why Not Use Posterior Sampling Instead of Injecting Noise?
>
> Thank you for raising this insightful question. In principle, posterior sampling from the VAE is a reasonable approach. However, in practice, we observe that the standard deviation of the VAE posterior tends to be extremely small, often around  ``1e-4`` level, due to the training dynamics and the ``tiny`` weighting of the KL divergence term. After all, pretrained AEs are mainly for reconstruction. So having a high entropy latent is not useful in their objectives.
> As a result, the injected noise is insufficient to effectively train the normalizing flow component of STARFlow. Therefore, we opt to introduce noise explicitly at a calibrated level to enable robust training.
> However, in the future, one can also explore options of training a tokenizer from scratch with native entropy regularization in the latent space for STARFlow.
>
> ---
>
> ## 4. T5-XL vs. T5-XXL Preference
>
> We used T5-XL as a representative encoder. In our preliminary experiments, we find T5XL has almost the same performance as T5XXL on text-conditioned generation while with significant small memory usage.
> Also, our goal was to demonstrate STARFlow's generality, and we anticipate similar outcomes with larger or alternative encoder backbones. We are open to further exploration in future work.
>
> ---

---

> > ### Comment · Reviewer_382B · 2025-08-02
> >
> > I thank the authors for their rebuttal.
> > Their answer on inclusion of performance metric and confirmation that the code will be open sourced make me keep my positive rating and I continue to argue for acceptance.

---

> > > ### Author Response · Authors · 2025-08-05
> > >
> > > Thank you very much for your support and we will include the analysis in the next version and code release!

---

### Official Review · Reviewer_M6cW · 2025-07-01

**Clarity:** 3
**Significance:** 4
**Originality:** 3
**Rating:** 5
**Confidence:** 5

**Summary:**

The paper presents STARFlow, a scalable generative model based on normalizing flows that achieves strong performance on high-resolution image synthesis. The core contributions include: 1) Deep-Shallow Architecture: A novel design where a deep Transformer block captures most of the model's capacity, followed by shallow blocks that are computationally cheap yet contribute non-negligibly. 2) Latent Space Learning: Training in the latent space of pretrained autoencoders, which proves far more effective than modeling pixels directly. 3) Novel Guidance Algorithm: A principled approach to classifier-free guidance that substantially improves sample quality, especially at high guidance weights. The model remains a single, end-to-end normalizing flow, allowing exact maximum likelihood training in continuous space without discretization. STARFlow achieves competitive results in both class- and text-conditional image generation, with sample quality approaching that of state-of-the-art diffusion models.

**Questions:**

Besides the weakness, I have following questions:
1. Joint latent-NF modeling: the authors mentioned the limitation of joint latent-NF model design. To my knowledge, existing latent-space based generative models leverage retrained auto encoders for compressing visual signals into latent codes, I wonder if there are better optimization solutions for joint latent-generative modeling?
2. Video modeling: any further plans to explore such model for video modeling? What's the main challenges for temporal modeling with your auto-regressive flow? Besides, finetuning the decoder for noised video decoding might leads to sub-optimal performance, especially for the motion modeling.

**Ethical Concerns:**

["NO or VERY MINOR ethics concerns only"]

**Final Justification:**

Most of my concerns are addressed during rebuttal, l keep my score as accept.

**Limitations:**

Overall, this paper successfully brings normalizing flows back to the generative modeling community and demonstrates its potential for scaling visual generation under high-resolution settings. This is a nice work, my major concerns and questions are shown in the weakness and questions.

**Quality:**

3

**Strengths And Weaknesses:**

**Strength:**
1. Theoretical Foundation: The paper establishes the theoretical universality of stacked autoregressive flows (AFs) for modeling continuous distributions, providing a solid basis for scaling up the normalizing flow models with autoregressive transformer architectures.
2. Improved Guidance: The proposed guidance algorithm offers a more principled approach to classifier-free guidance for AFs, leading to stable and high-quality image generation across a wide range of guidance weights.
3. Competitive Performance: STARFlow achieves results comparable to state-of-the-art diffusion and autoregressive models on both class-conditioned and text-to-image generation tasks, demonstrating the scalability and effectiveness of normalizing flows.
4. Well-presented and easy-to-follow: the technical insight of this paper is solid, and the overall presentation is well-structured and easy-to-follow.
5. Various applications: the proposed method could be further extended to downstream applications including image pinpointing and editing.


** Weaknesses:**
1.  Inference speed: Despite the deep-shallow architecture, the autoregressive nature of STARFlow might still results in relatively slow inference, especially for high-resolution images. It would be better to include some comparisons about the inference time and NFE.
2. Controllable generation: More controllable generation approaches such as ControlNet, T2I-Adapter enables user to produce images from various conditions, I wonder how the proposed method would perform when integrated more controllable signals to the system?
3. About decoder fine-tuning: finetuning the VAE decoder to handle noisy latents is an additional step that may not generalize well to other datasets or domains without further adaptation.

---

> ### Author Rebuttal · Authors · 2025-07-31
>
> We sincerely thank the reviewer for the encouraging evaluation and thoughtful suggestions. We are pleased that you found the theoretical foundation, guidance algorithm, competitive performance, and presentation of **STARFlow** to be strong. Below, we address each of your concerns and questions in detail.
>
> ---
>
> ## 1. Inference Speed and Comparison
>
> We appreciate your comments regarding inference speed. While STARFlow employs a deep-shallow separation to enhance efficiency, we agree that the autoregressive nature of the model may still lead to slower inference, especially at high resolutions. In response, We will provide a detailed comparison of inference time and FLOPs across baselines and STARFlow variants.
> Besides, there are also promising techniques which might be useful to improve the sampling speed such as Jacobi iteration. We will include these discussion in the updated version as well.
>
> ---
>
> ## 2. Controllable Generation
>
> Thank you for emphasizing the importance of controllability. We have included additional interactive editing results in Appendix Fig 10, and agree that control inputs like pose, depth, and sketch can be considered special cases of the broader editing framework we present. Our framework should be able to handle these editing cases as well with paired training set. More detailed experiments should be conducted to demonstrate. Please let us know if there are specific types of control signals we may have overlooked and will include them in the revised version.
>
> ---
>
> ## 3. Decoder Fine-tuning and Generalization
>
> We acknowledge the concern regarding decoder fine-tuning. In STARFlow, the VAE decoder is fine-tuned to better reconstruct from noisy latent codes, thereby improving sample quality. As noted, this approach may not generalize across datasets or domains. However, this limitation also points to an exciting research opportunity. In practice, we find that the fine-tuning process is **lightweight**, and only needs to be performed once per VAE, making it a practical and acceptable interim solution. In the future, we will also explore more end-to-end approaches without relying on finetuning the VAEs.
>
> ---
>
> ## 4. Joint Latent–NF Modeling
>
> As mentioned in our limitations, the joint design of latent space modeling and normalizing flows remains an open question. While we currently rely on decoder fine-tuning to bridge this gap, we agree that a principled joint training approach is a promising direction (especially for cases VAEs are hard to finetune like videos) for future work. We thank the reviewer for highlighting this important opportunity!
>
> ---
>
> ## 5. Temporal Extension for Video Modeling
>
> Thank you for the insightful suggestion on extending STARFlow to video modeling. We fully agree and it can be ``THE NEXT BIG STEP`` to show. This direction introduces both **challenges** and **opportunities**:
>
> The temporal coherence and causal nature of video data are naturally well-suited for modeling with autoregressive flows. However, this direction presents distinct challenges and opportunities:
>
> * **Challenge: Temporal Coherence vs. Latent Noise.** As you have rightly pointed out, introducing noise into the latent space can disrupt frame-to-frame coherence. Unlike in static images where visual fidelity is the primary goal, video generation demands strict temporal consistency. Adding noise can degrade this essential structure, leading to suboptimal performance and creating significant challenges for fine-tuning the underlying VAE.
> * **Opportunity: Causal and Interactive Prediction.** A further challenge, which we also view as a significant opportunity, is extending STARFlow to causal and interactive video prediction. Diffusion models are often suboptimal for such tasks due to their non-causal sampling process. STARFlow's autoregressive foundation may provides a more direct and potentially advantageous framework for handling causal generation.
>
> This tradeoff between modeling flexibility and temporal preservation highlights both the risks and promises of this research direction, which we are excited to explore in future extensions of our work.
>
> ---

---

> > ### Comment · Reviewer_M6cW · 2025-08-04
> >
> > I thanks the authros for their rebuttal. It is strongly suggested to include the experimental analysis in the next version and also release the code for the community. I continue to vote for acceptance.

---

> > > ### Author Response · Authors · 2025-08-05
> > >
> > > Thank you very much for your support and we will include the analysis in the next version and code release!

---

### Official Review · Reviewer_Fiwy · 2025-07-08

**Clarity:** 3
**Significance:** 3
**Originality:** 3
**Rating:** 6
**Confidence:** 5

**Summary:**

STARFLOW is an thorough extension/update from previous great work TARFLOW. In detail STARFLOW adds some practical training skills:
1. TARFLOW performs at pixel space of ImageNet-64/128, while STARFLOW is built in the re-designed VAE latent space.
2. TARFLOW can be seen as the part of the  shallow AF blocks in STARFLOW (the continuous visual “tokenizer”), while its deep block can leverage the pre-trained LLM.
3. The added noise (to pixel) skill in TARFLOW has been replaced as the 0.3 std noise added on VAE latents, and thus the extra denoising langevin dynamic step can be remved during inference by finetuning the decoder.
4. The deep-shallow block introduce the conditional flow by which  the conditions (like |sos| ) are only needed in  deep LLM block. Therefore the deep/shallow blocks are seperated for simplicity. Notice that the conditon in TARFLOW are added on the input to maintain the characteristics of Normalizing Flow.
5. New CFG are proposed.

Experiment parts:
1. ImageNet256/512 and text2img zero-shot COCO exps are provided, which shows the effectiveness of the proposed STARFLOW.
2. The editing applications and the visualization are impressive.

**Questions:**

See weakness:
1. The inference part should be described in detaill. I have not found how STARFLOW performs the inference with text and |sos| tokens except figure4. Please added the inference and sampling part in detail, especially how you deal with the |sos| token, and how you leverage the KV cache in inference.

2. The figure4 is hard to understand with various arrows. I know that    it is difficult to understand each steps in each coupling block of TARFLOW/STARFLOW. I consider that you can seperate the figure4 into four figures, the training/ inference of Deep blocks (multi-modal AR) and shallow blocks (visual encoder).


Questions:

3. The deep-shallow blocks are very similar with the design of Jetformer, even the   learning objective in final multi-modal AR is quite different. Could please claim the similarities and differences with  Jetformer?  Could the GMM in Jetformer replaced as the single N(0,1) in STARFLOW? or Could the  N(0,1) in STARFLOW be replaced with the  GMM in Jetformer?

4. In jetformer, when the input of txt2img pairs are replaced with the img2txt pairs, the model can be trained as a MLLM understanding model. Could STARFLOW be trained in the same way? I think STARFLOW could be improved as a unified model containing both generation and understanding with good potential. (Please don't show me the editing model in your paper

I am wiliing to raise my ratings if your solve all my questions.

**Ethical Concerns:**

["NO or VERY MINOR ethics concerns only"]

**Final Justification:**

The authors have solved most of my questions, beside the part of TARFLOWLM, which in my opinion is also a great work for NF modeling, even though the inference is unsatisfactory (which is the advance of JET NF network).
Another question is that can you provide some ablation insight for STARFLOW without VAE?


Considering about the novelty, completeness, and the current lack of diversity in CV generative  community,
I have raise my ratings from 4 to 6, and I strongly recomment AC to accept STARFLOW as an ORAL paper.

And the authors should  add the details in the final version, as they have promised, including:
1. Code

2. Add sampling and testing details.

3. Add Tarflow details with Ablation Insights.

4. Polish the figure.

5. Decalre the inference speed, espscially with the sequence of 32*32 =1024 (in Tarflow 256 * 256 case, for STARFLOW  case I don't know how you set your  VAE patch size)

6. Other reviewer's suggestions.

**Limitations:**

yes

**Quality:**

4

**Strengths And Weaknesses:**

Strengths:
1. Novelty:
Even though STARFLOW is an extension of TARFLOW, every improvement STARFLOW added to TARFLOW is highly practical, directly addressing the  bottlenecks of TARFLOW, and demonstrates good novelty. The design of deep-shallow blocks not only leverage the pre-trained LLM model, but also utilize the conditional flow to seperate the visual encoder (shallow AF blocks) and the multimodal "AR" part.
If we send text tokens into the shallow AF blocks (it contains about 64 layers transformer attn-ffn layers), the computation burden is very very heavy  and the train efficiency could be very very low. Besides, the text NF is very hard to train [2].

2. As for the rationality of deep-shallow blocks, STARFLOW provides sufficient proof.

3. The added VAE latent space not only reduce the dimension of pixel space, but also remove the reliance of pixel-added noise in TARFLOW which further requires an extra step (with lots of computation) to denoise. The added noise is transferring to the VAE latents, like the skills in Jetformer[1] (who added the latents noise (std=0.3 same with STARFLOW) on Normalizing Flow latents).


Neutral comment:
1. Performance:
Even though the performance is a bit lower than the mainstream diffusion or discrete AR methods,  in my consideration STARFLOW brings new light in current CV community and with the continuous improvement on Flow-Transformer-based models (other works like Jetformer[1]), the performance can catch up with DIT models  shortly afterwards.


[1] JETFORMER: AN AUTOREGRESSIVE GENERATIVE MODEL OF RAW IMAGES AND TEXT. ICLR 2025
[2] Flexible Language Modeling in Continuous Space with Transformer-based Autoregressive Flows. Arxiv 2025.07

Weakness:

1. The inference part should be described in detaill. I have not found how STARFLOW performs the inference with text and |sos| tokens except figure4. Please added the inference and sampling part in detail, especially how you deal with the |sos| token, and how you leverage the KV cache in inference.

2. The figure4 is hard to understand with various arrows. I know that    it is difficult to understand each steps in each coupling block of TARFLOW/STARFLOW. I consider that you can seperate the figure4 into four figures, the training/ inference of Deep blocks (multi-modal AR) and shallow blocks (visual encoder).

---

> ### Author Rebuttal · Authors · 2025-07-31
>
> We sincerely thank the reviewer for their thoughtful and constructive feedback. Below, we address each point in detail and outline the corresponding revisions we will incorporate in our next submission.
>
> ---
>
> # **Major Clarifications & Planned Revisions**
>
> ## **1. Inference with Text and `[SOS]` Tokens**
>
> We acknowledge that the inference pipeline could be explained more clearly. The inference (sampling) process corresponds to the **right plot** in Figure 4. Below is a step-by-step explanation of the sampling procedure:
>
> * Randomly sample $z_1, \ldots, z_N \sim \mathcal{N}(0, I^2)$.
> * For each block $t$ from $1$ to $T$:
>     * Initialize the KV cache for all attention layers.
>         * If $f_t$ is a deep block, we **prefill** the KV cache by performing a forward pass on $f_t$ with the T5 embeddings of the prompt (e.g., "A Corgi Dog") or one-hot vector embeddings of class labels.
>         * If $f_t$ is a shallow block, the KV cache is initialized with zeros.
>     * Initialize $x_0$ with the `[SOS]` token (a learnable continuous vector with the same dimension as the input channel size).
>     * For each step $n$ from $1$ to $N$:
>         * Run a forward pass $\mu_n, \sigma_n = f_t(x_{n-1}, \text{KV})$, updating the KV cache during the pass.
>         * If CFG is enabled and $f_t$ is a deep block, compute the updated $\mu_n, \sigma_n$ similarly to diffusion models but based on Proposition 2.
>         * Compute the next token $x_n = \mu_n + \sigma_n \cdot z_n$.
>     * The output sequence is reversed for the next block: $z_1, \ldots, z_N \leftarrow x_N, \ldots, x_1$.
> * Finally, reshape the sequence $x_1, \ldots, x_N$ to 2D and pass it to the VAE decoder to generate the final image.
>
> **Additional Clarifications:**
>
> * The equation in Figure 4 (right) computes the conditional mean and variance, which can be combined with the Classifier-Free Guidance (CFG) formulation from Proposition 2.
> * As an autoregressive model, we inject the `[SOS]` token (denoted `<s>` in Figure 4) as the first input during both training and inference for every autoregressive block, regardless deep or shallow blocks.
> * The KV cache works just like in standard LLMs, as each flow block is strictly autoregressive. In our implementation, the KV cache is used only at inference time. Each attention head has its own KV cache, which is appended with the current "key" and "value" at each step to compute attention based on the history.
> * Since conditioning is only performed in the deep blocks, their KV caches are longer than those of shallow blocks. CFG is also applied only in the deep blocks.
>
> *In the revision, we will add a detailed explanation to explicitly elaborate on this inference procedure.*
>
> ---
>
> ## **2. Complexity of Figure 4**
>
> We appreciate the reviewer’s suggestion regarding Figure 4. We agree that breaking it into four modular sub-figures would enhance clarity.
>
> *We will update the figure accordingly in the revised manuscript.*
>
> ---
>
> ## **3. Clarification and Relationship to JetFormer**
>
> We will revise the text to clearly differentiate STARFlow from JetFormer and clarify our design choices in greater detail. The major differences are as follows:
>
> * **JetFormer's Architecture:** JetFormer uses a flow model called Jet (a Transformer-based RealNVP) to map pixels to a latent space. It employs a GMM-based latent prior and is not fully invertible. Therefore, JetFormer is technically a VAE that optimizes the ELBO. In our experiments, we found that the Jet flow requires many blocks to be sufficiently expressive.
> * **STARFlow's Architecture:** In STARFlow, all layers are affine autoregressive flows and are implemented uniformly. The only difference is that deep blocks have more layers and include text conditioning. This means our flow model directly optimizes the likelihood and is fully end-to-end trainable, and the whole network is invertible. As our Prop.1 shows, only 2-3 autoregressive flow blocks are needed to capture an arbitrary distribution, which makes the model highly scalable. This is a key difference from JetFormer.
> * **Operating Space:** JetFormer operates directly on pixels, whereas our work operates in the latent space of a pretrained auto-encoder, which is a more scalable approach. In this sense, the overall objective of STARFlow is also optimizing an ELBO, but with a fixed encoder. So there is no ``moving target``.
> * **Factoring-Out:** JetFormer uses a multi-scale factor-out technique to compress the space for its GMM prior. While STARFlow does not currently use this, we consider it a promising future extension.
> * **Replace N(0,I) with GMM:** STARFlow could replace the deep block with a GMM prior, but this is not necessary as the model is already powerful enough. **A key drawback of the GMM prior is the lack of an easy analytic solution for guidance**, requiring approximation via rejection sampling. In STARFlow, the Gaussian prior allows us to use a closed-form guidance (Proposition 2), similar to diffusion models, which is a significant advantage. *
> * **Replace GMM with N(0,I):** JetFormer **cannot** simply replace its GMM with a Gaussian, as this would result in insufficient modeling power from the Jet flow alone (by checking results from Jet's paper).
>
> ---
>
> ## **4. Potential Use as a Unified Model**
>
> * **Short Answer: Yes**
> * STARFlow can be extended for joint modeling with a language modeling loss (cross-entropy) to perform both understanding and generation tasks. By incorporating a joint modeling objective, STARFlow could, *in theory*, work as well as other unified models like Transfusion, which can perform text-to-image generation, editing, and, in the reverse direction, image understanding.
> * **The Challenges:** However, to train a state-of-the-art unified model, we would need to consider how to incorporate information from a ViT encoder for understanding tasks, which is non-trivial. The critical question is whether STARFlow itself should be trained to "encode".
>     * **If not:** We could train an MLLM version of STARFlow. As supporting evidence, in our paper, we successfully finetuned a pretrained Gemma for text-to-image generation. In this setup, a separate text encoder is not needed, and it achieves image generation quality nearly as good as the default version. We could further explore approaches where the MLLM (with a ViT encoder) is frozen, fixing the understanding ability, though this may not be a "truly unified" model.
>     * **If yes:** The challenge would be training STARFlow to match the performance of a dedicated ViT. Recent work like Show-o2 has made attempts in this direction.
> * In summary, STARFlow can be extended for unified models, and our preliminary results are suggestive of this potential. However, significant research and exploration would be needed to make it performant.
>
> ---
>
> # **Summary of Revisions**
>
> To summarize, we will:
>
> 1.  **Clarify the inference process,** including the role of `[SOS]` tokens and KV caching.
> 2.  **Redesign Figure 4** to enhance modular understanding and readability.
> 3.  **Differentiate STARFlow from JetFormer** with clear explanations and a comparative summary.
> 4.  **Refine the manuscript’s presentation,** improving clarity, labeling, and overall readability.
>
> We hope these changes will address the reviewer’s concerns and significantly enhance the clarity and impact of our paper.

---

> > ### Author Response · Authors · 2025-08-05
> >
> > Dear Reviewer,
> >
> > Thank you very much again for reviewing our submission! We’ve carefully addressed your comments in the rebuttal. If there are any remaining questions or concerns, we would greatly appreciate to further clarify before the discussion phase ends.
> >
> > Thank you again for your thoughtful consideration!

---

> > > ### Comment · Reviewer_Fiwy · 2025-08-05
> > >
> > > Thanks for your valuable and detailed feedbacks about STARFLOW. The feedbacks have solved most of my questions.
> > > And (S)TARFLOW series are really  great works for current CV community which is facing a serious issue of homogenization.
> > >
> > > I have another question about the performance of  the upgraded TARFLOW. Could please provide the details about the configs in Table 1? Have you utilized VAE for the upgraded TARFLOW performance?
> > >
> > > Besides, I am looking forward to seeing the unify model by STARFLOW  and TARFLOWLM, which might be powerful to sovle the confilcts with the generation loss and the understanding loss.
> > >
> > > And considering about other reviewers' comments, hope that you can release the code public.

---

> ### Author Response · Authors · 2025-08-05
> **Response for additional question**
>
> Thank you for the thoughtful response. We are glad to hear that our earlier rebuttal has resolved **most** of your questions.
>
> ## Reviewer Comment
> > **Q1:** Could you provide the configuration details for the models in Table 1?
> > **Q2:** Did you employ a VAE to obtain the upgraded TARFlow performance?
>
> ## Our Response
>
> ### Table 1: Model Summary
>
> | Model | FID&nbsp;↓ | Params |
> |-------|-----------|--------|
> | **TARFlow** (Zhai *et al.* 2024)¹ | 5.56 | 1.3 B |
> | **TARFlow + deep-shallow** | 4.69 | 1.4 B |
> | **STARFlow (Ours)** | **2.40** | 1.4 B |
>
> ¹ We re-implemented TARFlow using the authors’ public code, training at **256 × 256** on ImageNet—​a setting not reported in the original paper.
>
> ---
>
> ### Implementation Details
>
> | Variant | Architecture | Patch Size | Space |
> |---------|--------------|-----------|-------|
> | **Baseline TARFlow** | 8 AR blocks × 8 layers, dim=1280 | 8×8 | Pixel |
> | **TARFlow + deep-shallow** | 1 deep block (18 layers) + 5 shallow blocks (2 layers), dim=2048 | 8×8 | Pixel |
> | **STARFlow** | Same deep-shallow backbone, dim=2048 | 1x1 | **SD-VAE** |
>
> All three models were trained under comparable computational budgets, and the inference cost is almost the same.
>
> ---
>
> ### Ablation Insights
> ```
> standard TARFlow (pixel) <  deep-shallow TARFlow (pixel) <  standard TARFlow (latent)* <  STARFlow architecture (deep-shallow model + latent NF)
> ```
>
> \* Early experiments showed that simply shifting **standard** TARFlow to latent space still lagged behind our deep-shallow design. We will add this datapoint in the next revision for completeness.
>
> ---
>
> ### On VAE Usage
>
> No VAE was employed in any TARFlow variant. Performance gains come from:
>
> 1. **Deep-shallow hierarchy** (better capacity allocation)
> 2. **Latent normalizing flows** (more compact space + VAE decoder denoising)
>
> ---
>
> ### Code & Checkpoints
>
> Sorry for making the wait. We are finalizing the code and expect to make it public shortly.
> As we want to release the associated text-to-image checkpoints, it takes a bit of time to prepare for review.
>
> ---
>
> If anything remains unclear or requires further details, please let us know—​we are happy to help as soon as we can!
>
> Best regards,

---

> ### Comment · Reviewer_Fiwy · 2025-08-05
> **Rasing my ratings from 4 to 6 and Recommend to ORAL paper**
>
> The authors have solved most of my questions, beside the part of TARFLOWLM, which in my opinion is also a great work for NF modeling, even though the inference is unsatisfactory (which is the advance of JET NF network).
> Another question is that can you provide some ablation insight for STARFLOW without VAE?
>
>
> Considering about the novelty, completeness, and the current lack of diversity in CV generative  community,
> I have raise my ratings from 4 to 6, and I strongly recomment AC to accept STARFLOW as an ORAL paper.
>
> And the authors should  add the details in the final version, as they have promised, including:
> 1. Code
> 2. Add sampling and testing details.
> 3. Add Tarflow details with Ablation Insights.
> 4. Polish the figure
> 5. Decalre the inference speed, espscially with the sequence of 32*32 =1024 (in Tarflow 256 * 256 case, for STARFLOW  case I don't know how you set your  VAE patch size)
> 6. Other reviewer's suggestions.

---

> > ### Author Response · Authors · 2025-08-05
> >
> > Thank you very much for your support. We will include the requested details in the revision!

---

### Decision · Program_Chairs · 2025-09-17

**Decision:**

Accept (spotlight)

**Comment:**

The paper presents STARFlow, an autoregressive flow (a type of normalizing flow) for high-resolution image synthesis.
It builds on TARFlow and introduces key architectural and algorithmic changes to improve scalability, including a deep–shallow design and feature-space learning.

Reviewers unanimously praised the work for its novelty, competitive performance, and extensive experiments, and recommended acceptance as of broad interest to the computer vision community.

For the camera-ready, as requested by the reviewers and acknowledged by the authors, please report detailed training and inference efficiency/cost metrics and release the source code.